# Federated ADMM from Bayesian Duality

**Thomas Möllenhoff**[*]
RIKEN Center for AI Project
Tokyo, Japan
thomas.moellenhoff@riken.jp

**Siddharth Swaroop**[*]
University College London
London, United Kingdom
s.swaroop@ucl.ac.uk

**Finale Doshi-Velez**
Harvard University
Cambridge, United States
finale@seas.harvard.edu

**Mohammad Emtiyaz Khan**[†]
RIKEN Center for AI Project
Tokyo, Japan
emtiyaz.khan@riken.jp

## Abstract

We propose a new Bayesian approach to generalize the federated Alternating Direction Method of Multipliers (ADMM). We show that the solutions of variational-Bayesian (VB) objectives are associated with a duality structure that not only resembles the structure of ADMM's fixed-points but also generalizes it. For example, ADMM-like updates are recovered when the VB objective is optimized over the isotropic-Gaussian family, and new non-trivial extensions are obtained for other exponential-family distributions. These extensions include a Newton-like variant that converges in one step on quadratic objectives and an Adam-like variant that yields up to 7% accuracy boosts for deep heterogeneous cases. Our work opens a new Bayesian way to generalize ADMM and other primal-dual methods.

## 1 Introduction

The Alternating Direction Method of Multipliers (ADMM) forms the backbone of many federated learning algorithms (Acar et al., 2021; Gong et al., 2022; Mishchenko et al., 2022; Wang et al., 2022; Zhang et al., 2021; Zhou & Li, 2023). The goal of federated learning is to train a *global* model at a server without ever accessing the *local* data stored at the clients. In ADMM, this is achieved through communication between the server and clients, as shown in Fig. 1 (left). The server first broadcasts the global parameter to the clients, and the clients use it to update their local parameters. The clients then send the updated parameters and their local gradients back to the global server, where they are combined to update the global parameter. By repeating these steps, ADMM can recover the solution obtained by jointly training on all data, provably so in many cases.

ADMM was proposed back in the 1970s (Gabay & Mercier, 1976; Glowinski & Marroco, 1975) and its roots can be traced back to an early work by Douglas & Rachford (1956). Yet, it continues to be used more or less in the same form it was originally proposed. The robustness of ADMM's algorithmic structure is intriguing, and we wonder whether there is a more general formulation of this structure. Such formulations could be especially relevant to tackle new issues that arise in federated deep learning. Our search for new generalizations follows a recent result by Swaroop et al. (2025) who connect a variational-Bayesian (VB) version of federated learning to ADMM. Their work shows a close resemblance between ADMM and VB, but it falls short of deriving ADMM as a special case of VB. Our goal here is to fill this gap by proposing a new Bayesian way to generalize ADMM.

We propose a general VB framework that can be used to derive and extend federated ADMM. Our main result is to show that the solutions of the VB objective are associated with a duality structure that not only resembles ADMM's fixed-point equations, but also naturally generalizes them. We call this structure Bayesian-duality and use it to propose a new algorithm called Bayesian-ADMM. Our generalization is obtained by making two changes to ADMM. First, we introduce distributions over parameters and, second, we replace gradients by natural gradients (Fig. 1). The use of natural gradients is crucial to fix the issue of Swaroop et al. (2025). It enables us to derive classical ADMM

---

[*]Equal contribution, [†]Corresponding author. Code is available here.

**ADMM**

**BayesADMM**

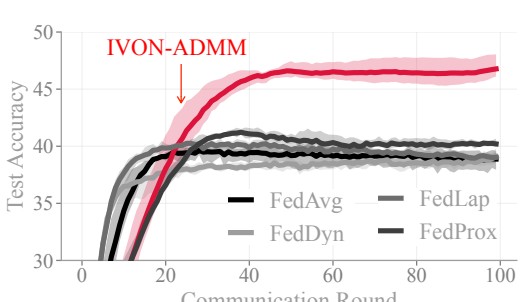

Figure 1: In ADMM, the server broadcasts the global parameter $\boldsymbol{\theta}_g$ to the clients who then update their local $\boldsymbol{\theta}_k$ and send them back to the server along with the gradients $\nabla\ell_k$ of their loss functions. Bayesian-ADMM generalizes ADMM by using distributions over global and local parameters (denoted by $q_g$ and $q_k$, respectively) and replacing gradients by natural gradients (denoted by $\widetilde{\nabla}\mathbb{E}[\ell_k]$).

**ResNet-20 on CIFAR-100 with 10 clients**

**MLP on MNIST with 100 clients**

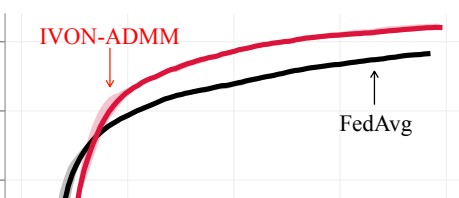

Figure 2: We derive an Adam-like extension of ADMM, called IVON-ADMM, which gives up to 7% accuracy boost over existing federated deep-learning methods (left) without any increase in the cost and overall runtime (right). Details of the experiment are in Sec. 4.

as a special case of Bayesian-ADMM that employs isotropic Gaussian posteriors. New non-trivial generalizations are also automatically obtained by using other exponential-family distributions.

We derive two new extensions of ADMM using Bayesian-ADMM. The first extension is a Newton-like variant obtained by using multivariate Gaussian distributions. Unlike classical ADMM, this new variant converges in one communication round when applied to quadratic objectives. The second extension is an Adam-like variant obtained by restricting the covariance to be a diagonal matrix. This variant can be efficiently implemented by using the IVON optimizer of Shen et al. (2024) and yields up to 7% accuracy boost for deep heterogeneous cases without increasing the cost and overall runtime (Fig. 2). Ultimately, our work opens a new Bayesian way to generalize ADMM and other primal-dual methods. This was not possible before, even though a lot of work has been done recently on Bayesian methods for federated deep learning (Al-Shedivat et al., 2021; Guo et al., 2023; Kotelevskii et al., 2022; Louizos et al., 2021; Pal et al., 2024; Yurochkin et al., 2019).

## 2 FEDERATED LEARNING VIA ADMM

Federated learning aims to train a global model with parameter $\boldsymbol{\theta}_g$ at a central server by communicating with $K$ clients and without ever gaining access to their local data. The clients can access their data through local loss functions denoted by $\ell_k(\boldsymbol{\theta})$ for the $k$'th client. The goal is to solve for

$$\boldsymbol{\theta}_g^* = \arg\min_{\boldsymbol{\theta}} \sum_{k=0}^{K} \ell_k(\boldsymbol{\theta}), \tag{1}$$

where $\ell_0(\boldsymbol{\theta})$ denotes the regularizer. However, since the server is not allowed direct access to $\ell_k$, it is forced to solve this problem by communicating with clients. The main idea is to perform distributed

**ADMM's Duality**     **Bayesian Duality**

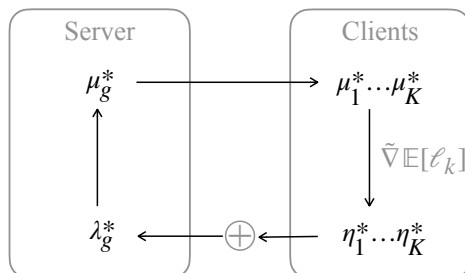

Figure 3: The left side shows the dual structure associated with ADMM's fixed-point equation (Eq. 4). The right side shows our new Bayesian-Duality structure associated with VB's optimality condition (Eq. 13). In ADMM, $\boldsymbol{\theta}_g^*$ and $\boldsymbol{\theta}_k^*$ are the primal variables, while $\mathbf{v}_g^*$ and $\mathbf{v}_k^*$ are dual variables. Analogously, in Bayesian Duality, $\boldsymbol{\mu}_g^*$ and $\boldsymbol{\mu}_k^*$ are primal, while $\boldsymbol{\lambda}_g^*$ and $\boldsymbol{\eta}_k^*$ are dual.

optimization by exploiting the structure of the solution. For instance, for $\ell_0(\boldsymbol{\theta}_g) = \frac{1}{2}\|\boldsymbol{\theta}_g\|^2$, the optimality condition of the problem can be written as a sum over local gradients (proof in App. A),

$$\boldsymbol{\theta}_g^* = -\sum_{k=1}^{K} \nabla\ell_k(\boldsymbol{\theta}_g^*). \tag{2}$$

This structure suggests distributing the computation of local gradient $\nabla\ell_k$ across the clients, and gathering those results to estimate $\boldsymbol{\theta}_g^*$. ADMM provides a framework to perform such distributed optimization. It introduces local $\boldsymbol{\theta}_k$ and aims to solve an equivalent constrained optimization problem:

$$\min_{\boldsymbol{\theta}_g, \boldsymbol{\theta}_{1:K}} \sum_{k=1}^{K} \ell_k(\boldsymbol{\theta}_k) + \ell_0(\boldsymbol{\theta}_g), \text{ such that } \boldsymbol{\theta}_g = \boldsymbol{\theta}_k \text{ for all } k = 1, 2, \ldots, K. \tag{3}$$

The problem can be solved by formulating a Lagrangian with multipliers, denoted by $\mathbf{v}_k$ for the $k$'th constraint. The stationarity condition of the Lagrangian, shown below, then provides a way to distribute the computations (proof in App. A),

$$\boldsymbol{\theta}_k^* = \boldsymbol{\theta}_g^*, \qquad \mathbf{v}_k^* = -\nabla\ell_k(\boldsymbol{\theta}_k^*), \qquad \mathbf{v}_g^* = \sum_{k=1}^{K} \mathbf{v}_k^*, \qquad \boldsymbol{\theta}_g^* = \mathbf{v}_g^*. \tag{4}$$

The first condition says that we set $\boldsymbol{\theta}_k^* = \boldsymbol{\theta}_g^*$ to satisfy the constraint. The second condition sets the optimal multiplier $\mathbf{v}_k^*$ to the negative of the local gradient. The third condition gathers all the multipliers into a global variable $\mathbf{v}_g^*$, and finally the fourth condition uses it to get the global $\boldsymbol{\theta}_g^*$ back.

The optimality condition can be drawn in the form of a *dual structure*, similarly to that used by Rockafellar (1967, Fig. 2). The parameters $\boldsymbol{\theta}_g^*$ and $\boldsymbol{\theta}_k^*$ are the primal variables defined in the space of valid parameters, while $\mathbf{v}_k^*$ and $\mathbf{v}_g^*$ are dual variables defined in the space of valid gradients. The duality structure, shown in the left panel of Fig. 3, summarizes the flow of information. First, the server broadcasts its global $\boldsymbol{\theta}_g^*$ to the clients and the clients set $\boldsymbol{\theta}_k^* = \boldsymbol{\theta}_g^*$. Then, the gradient $\nabla\ell_k(\boldsymbol{\theta}_k^*)$ is evaluated and assigned to the dual $\mathbf{v}_k^*$. All the dual variables are then gathered using a sum and assigned to $\mathbf{v}_g^*$, which is then used to obtain $\boldsymbol{\theta}_g^*$.

The Lagrangian also yields an algorithm to perform optimization that preserves this flow of information. Essentially, we first perform local optimizations at all clients to obtain $\boldsymbol{\theta}_k$, then we update the dual $\mathbf{v}_k$, and finally these results are communicated to the server which updates $\boldsymbol{\theta}_g$. A detailed derivation is in App. A and the resulting updates are summarized below:

$$\text{Client updates:} \quad \boldsymbol{\theta}_k \leftarrow \arg\min_{\boldsymbol{\theta}_k} \ell_k(\boldsymbol{\theta}_k) + \mathbf{v}_k^\top \boldsymbol{\theta}_k + \frac{\rho}{2}\|\boldsymbol{\theta}_k - \boldsymbol{\theta}_g\|^2, \tag{5}$$

$$\mathbf{v}_k \leftarrow \mathbf{v}_k + \rho(\boldsymbol{\theta}_k - \boldsymbol{\theta}_g), \tag{6}$$

$$\text{Server update:} \quad \boldsymbol{\theta}_g \leftarrow \arg\min_{\boldsymbol{\theta}_g} \ell_0(\boldsymbol{\theta}_g) - \sum_{k=1}^{K} \mathbf{v}_k^\top \boldsymbol{\theta}_g + \sum_{k=1}^{K} \frac{\rho}{2}\|\boldsymbol{\theta}_g - \boldsymbol{\theta}_k\|^2. \tag{7}$$

Here, $\rho > 0$ is the (inverse) step-size, added to handle the quadratic proximal terms for each update.

An important property of these updates, as shown in App. A, is that after any client's updates, the dual vectors are simply equal to their negative local gradients, that is,

$$\mathbf{v}_k = -\nabla \ell_k(\boldsymbol{\theta}_k). \tag{8}$$

As a result, $\mathbf{v}_g$ slowly approaches the optimal $\boldsymbol{\theta}_g^*$. This can be more clearly seen for the special case when $\ell_0(\boldsymbol{\theta}_g) = \frac{1}{2}\|\boldsymbol{\theta}_g\|^2$, where the update for $\boldsymbol{\theta}_g$ simplifies nicely (see Eq. 27 in App. A).

The form of the ADMM updates has remained more or less the same since its original proposal in the 1970s. For instance, variants to accelerate ADMM's convergence only introduce additional variables, but do not change the form of the algorithm; this includes methods that use overrelaxation (Eckstein & Bertsekas, 1992) or momentum (Chambolle & Pock, 2016). The same is true of variants that replace quadratic proximal terms by scaled (Mahalanobis) norms (Goldstein et al., 2013; Pock & Chambolle, 2011; Ye et al., 2020) or Bregman divergences (Ma et al., 2025; Wang et al., 2014; Wang & Banerjee, 2014). The robustness of the ADMM's algorithmic structure is intriguing and we wonder whether there is a more general formulation to go beyond it. Such generalizations could be useful to handle new issues arising in federated deep learning due to client-heterogeneity and missing data.

Our approach here is inspired by the work of Swaroop et al. (2025) who show a connection between federated ADMM and a distributed variational Bayesian (VB) method called Partitioned Variational Inference (PVI) (Ashman et al., 2022). They find a line-by-line correspondence between the ADMM and PVI updates, but fall short of establishing an exact connection. In what follows, we will fix this issue by using a duality structure associated with the solutions of the VB objective.

## 3 BAYESIAN DUALITY TO GENERALIZE ADMM

We now present a Bayesian-duality structure to generalize ADMM. We first introduce the dual structure of the VB fixed-point equation, then use it to define Bayesian duality and Bayesian-ADMM.

### 3.1 VARIATIONAL BAYES AND ITS SOLUTIONS

We will use a VB reformulation of Eq. 1 which 'lifts' the original problem to define a new problem where we optimize with respect to probability distributions $q$ in a set $\mathcal{Q}$. The goal then is to solve for

$$q_g^* = \operatorname*{argmin}_{q \in \mathcal{Q}} \sum_{k=1}^K \underbrace{\mathbb{E}_q[\ell_k]}_{=\mathcal{L}_k} + \mathrm{KL}(q \,\|\, \pi_0). \tag{9}$$

The second term is the KL divergence between the distribution $q$ and prior $\pi_0 \propto \exp(-\ell_0)$. When $\ell_k$ corresponds to proper likelihoods and $\mathcal{Q}$ contains all possible distributions, then $q_g^*$ is the standard posterior distribution. Restricting $\mathcal{Q}$ to a smaller set, say a set of Gaussian distributions, yields an approximation to the posterior. An important property of the VB formulation is that the solution $\boldsymbol{\theta}_g^*$ of Eq. 1 can be recovered from $q_g^*$ when $\mathcal{Q}$ is set to isotropic Gaussians (Khan & Rue, 2023, App. C.1). We will use this technique later to derive ADMM from a federated algorithm for the VB objective.

The first element needed to generalize ADMM is to set $q \in \mathcal{Q}$ to be of exponential-family (EF) form. An EF has a *log*-linear form with respect to a sufficient statistic, denoted by $\mathbf{T}(\boldsymbol{\theta})$, shown below,

$$q(\boldsymbol{\theta}) = h(\boldsymbol{\theta}) \exp\left(\langle \boldsymbol{\lambda}, \mathbf{T}(\boldsymbol{\theta}) \rangle - A(\boldsymbol{\lambda})\right). \tag{10}$$

Here, $\boldsymbol{\lambda}$ denotes the natural parameter, $h(\boldsymbol{\theta})$ denotes the base measure, and $A(\boldsymbol{\lambda})$ is the (convex) log-partition function. For readers unfamiliar with EFs, a brief introduction is included in App. B.1.

The second element is to define natural gradients that scale the gradients by using the inverse Fisher-information matrix of $q$. For EF distributions, the natural gradient can be conveniently obtained by using the expectation parameter, defined as $\boldsymbol{\mu} = \nabla A(\boldsymbol{\lambda}) = \mathbb{E}_q[\mathbf{T}(\boldsymbol{\theta})]$. We can show that gradients with respect to $\boldsymbol{\mu}$ are equal to natural gradients with respect to $\boldsymbol{\lambda}$ (Khan & Rue, 2023, Eq. 4):

$$\nabla_{\boldsymbol{\mu}} \mathcal{L}(\boldsymbol{\mu}) = \mathbf{F}(\boldsymbol{\lambda})^{-1} \nabla_{\boldsymbol{\lambda}} \mathcal{L}(\boldsymbol{\mu}(\boldsymbol{\lambda})) = \widetilde{\nabla}_{\boldsymbol{\lambda}} \mathcal{L}(\boldsymbol{\mu}(\boldsymbol{\lambda})), \tag{11}$$

where $\mathbf{F}(\boldsymbol{\lambda})$ and $\widetilde{\nabla}_{\boldsymbol{\lambda}}$ denote the Fisher information matrix and natural gradient with respect to $\boldsymbol{\lambda}$, respectively. This equation holds because the $(\boldsymbol{\lambda}, \boldsymbol{\mu})$ pair constitutes a *dual-map* (Amari, 2016)

through the convex function $A(\boldsymbol{\lambda})$ and its Fenchel conjugate $A^*(\boldsymbol{\mu})$. Such maps are common in information geometry and convex optimization (Malagò et al., 2011; Raskutti & Mukherjee, 2015). In the rest of the paper, we will drop the subscript $\boldsymbol{\mu}$ and write the natural gradient as $\nabla\mathcal{L}(\boldsymbol{\mu})$.

Khan & Rue (2023, Eq. 5) used this property to express the solution $q_g^*$ in terms of natural gradients. Specifically, they show that, when $h(\boldsymbol{\theta})$ is a constant, the natural parameter $\boldsymbol{\lambda}_g^*$ of $q_g^*$ is given by

$$\boldsymbol{\lambda}_g^* = -\sum_{k=0}^{K} \nabla\mathcal{L}_k(\boldsymbol{\mu}_g^*) \qquad \Longrightarrow \qquad q_g^*(\boldsymbol{\theta}) = \frac{1}{\mathcal{Z}^*} \prod_{k=0}^{K} t_k^*(\boldsymbol{\theta}), \tag{12}$$

where we define the *optimal* 'site' functions as $t_k^*(\boldsymbol{\theta}) = \exp\left(-\langle\nabla\mathcal{L}_k(\boldsymbol{\mu}_g^*), \mathbf{T}(\boldsymbol{\theta})\rangle\right)$, and $\mathcal{Z}^*$ is the partition function. A proof is included in App. B.2. The two equations were originally proposed in Khan & Lin (2017, Eq. 11) and Khan & Nielsen (2018, Eq. 18), respectively.

These equations have a strikingly similar form to Eq. 2. There, $\boldsymbol{\theta}_g^*$ is equal to the sum of the local gradients, while here, $\boldsymbol{\lambda}_g^*$ is equal to the sum of local *natural* gradients and $q_g^*$ is equal to the product of the local site functions. The existence of these equations is also mentioned in Swaroop et al. (2025, Eq. 4), but they do not make use of natural gradients. We will now show that, by using these equations, we can directly generalize ADMM's dual structure.

## 3.2 BAYESIAN DUALITY

We will now derive a dual structure for the VB solutions by drawing an analogy to Eq. 4. There, we introduce local $\boldsymbol{\theta}_k^*$ and their corresponding dual variables $\mathbf{v}_k^*$ which are set to the (negative) local gradients $\nabla\ell_k(\boldsymbol{\theta}_k^*)$. The $\mathbf{v}_k^*$ are then added together to obtain $\mathbf{v}_g^*$ and $\boldsymbol{\theta}_g^*$. For the VB fixed point, we follow the same process to define local variables and their corresponding dual variables.

We introduce local distributions $q_k^*$ with pair $(\boldsymbol{\lambda}_k^*, \boldsymbol{\mu}_k^*)$ and denote their corresponding dual variables by $\boldsymbol{\eta}_k^*$. Using these variables, we can express the optimality condition in Eq. 12 as the following equivalent set of conditions:

$$\boldsymbol{\mu}_k^* = \boldsymbol{\mu}_g^*, \qquad \boldsymbol{\eta}_k^* = -\nabla\mathcal{L}_k(\boldsymbol{\mu}_k^*), \qquad \boldsymbol{\lambda}_g^* = \sum_{k=0}^{K} \boldsymbol{\eta}_k^*, \qquad \boldsymbol{\mu}_g^* = \nabla A(\boldsymbol{\lambda}_g^*). \tag{13}$$

Here, $\boldsymbol{\mu}_g^*$ and $\boldsymbol{\mu}_k^*$ are the primal variables in the expectation-parameter space, while $\boldsymbol{\eta}_k^*$ and $\boldsymbol{\lambda}_g^*$ are the dual variables in the space of valid natural gradients. The first three conditions are a direct extension of those in Eq. 4. The last condition connects the pair $(\boldsymbol{\lambda}_g^*, \boldsymbol{\mu}_g^*)$ through the dual map of the EF. This duality of EFs provides a foundation to build the duality of VB, which we refer to as the Bayesian-duality structure. The dual structure is visualized in the right panel in Fig. 3.

This duality is missed in the work of Swaroop et al. (2025) who connect VB to ADMM without using natural gradients. They show that the parameters of sites $t_k$ correspond to the dual variables $\mathbf{v}_k$ in ADMM, but fall short of providing an exact correspondence. This can be fixed by using Bayesian duality to write a dual structure in the space of distributions and sites, as shown below:

$$q_k^* = q_g^*, \qquad t_k^*(\boldsymbol{\theta}) = \exp\left(-\langle\widetilde{\nabla}\mathbb{E}_{q_k^*}[\ell_k], \mathbf{T}(\boldsymbol{\theta})\rangle\right), \qquad t_g^* = \prod_{k=0}^{K} t_k^*, \qquad q_g^* = \frac{t_g^*}{\mathcal{Z}^*}. \tag{14}$$

This shows that the sites are locally constructed using natural gradients. These are then multiplied together and normalized to get the global $q_g^*$ which is also equal to the local $q_k^*$. The normalization in the last step plays the same role as EF's dual map in Eq. 13.

## 3.3 BAYESIAN-ADMM

We will now present an optimization algorithm that follows the flow of information suggested by the Bayesian Duality structure. The algorithm closely follows the ADMM updates given in Eqs. 5 to 7, but makes two important modifications. First, the quadratic proximal terms are replaced by the KL divergence which is a more natural choice for the EFs. Second, the update of the dual $\boldsymbol{\eta}_k$ is modified to ensure a similar condition to Eq. 8 holds, that is, we make sure that the duals are set to the latest local natural gradients after every local update.

The final algorithm, which we refer to as Bayesian-ADMM, is shown below:

$$\text{Client updates:} \quad \boldsymbol{\mu}_k \leftarrow \arg\min_{\boldsymbol{\mu}_k} \; \mathcal{L}_k(\boldsymbol{\mu}_k) + \langle \boldsymbol{\eta}_k, \boldsymbol{\mu}_k \rangle + \rho \text{KL}(q_k \, \| \, q_g) \tag{15}$$

$$\boldsymbol{\eta}_k \leftarrow \boldsymbol{\eta}_k + \rho(\boldsymbol{\lambda}_k - \boldsymbol{\lambda}_g) \tag{16}$$

$$\text{Server update:} \quad \boldsymbol{\mu}_g \leftarrow \arg\min_{\boldsymbol{\mu}_g} \; \text{KL}(q_g \, \| \, \pi_0) - \sum_{k=1}^{K} \langle \boldsymbol{\eta}_k, \boldsymbol{\mu}_g \rangle + \sum_{k=1}^{K} \rho \text{KL}(q_g \, \| \, q_k). \tag{17}$$

The first and third lines can both be seen as modified VB problems that are run at the clients and server respectively. The modification in the second line deserves a bit more explanation. If we follow the update in Eq. 6, then the dual update should be $\boldsymbol{\eta}_k \leftarrow \boldsymbol{\eta}_k + \rho(\boldsymbol{\mu}_k - \boldsymbol{\mu}_g)$. However, this update does not ensure that $\boldsymbol{\eta}_k$ are equal to $\nabla \mathcal{L}_k(\boldsymbol{\mu}_k)$. Instead, if we use the difference between natural parameters $\boldsymbol{\lambda}_k - \boldsymbol{\lambda}_g$, this issue is resolved; see a proof in App. C.1. Subtraction of natural parameters corresponds to division in EFs, but arithmetic in expectation parameters has no such equivalence. We discuss the dual update in App. C.2, along with its relationship to Bregman ADMM (Wang & Banerjee, 2014) and the Bayesian learning rule (Khan & Rue, 2023).

The algorithm can also be written in a distributional form by defining sites $t_k = \exp(\langle \boldsymbol{\eta}_k, \mathbf{T}(\boldsymbol{\theta}) \rangle)$,

$$\text{Client updates:} \quad q_k \leftarrow \arg\min_{q_k} \; \mathbb{E}_{q_k}[\ell_k + \log t_k] + \rho \text{KL}(q_k \, \| \, q_g) \tag{18}$$

$$t_k \leftarrow t_k \left( \frac{q_k}{q_g} \right)^{\rho} \tag{19}$$

$$\text{Server update:} \quad q_g \leftarrow \arg\min_{q_g} \; \text{KL}(q_g \, \| \, \pi_0) - \sum_{k=1}^{K} \mathbb{E}_{q_g}[\log t_k] + \sum_{k=1}^{K} \rho \text{KL}(q_g \, \| \, q_k). \tag{20}$$

This algorithm closely resembles the PVI algorithm used by Swaroop et al. (2025). Our derivation here highlights an additional fact that the site parameters $\boldsymbol{\eta}_k$ are equal to the negative local natural gradients after every local update. The main differences to PVI are highlighted in red, which include the use of step-size $\rho$ and addition of the KL term in the global update. In fact, PVI is more similar to an algorithm by Tseng (1991) called the Alternating Minimization Algorithm (AMA). The importance of step-sizes are well-established for AMA (see Tseng (1991, Eq. 3.4d)), and this connection can be used to improve the convergence of PVI as well. We show such results later in Fig. 5b. More details regarding the differences between Bayesian-ADMM and PVI are given in App. C.3.

We note that our derivation here is to follow the ADMM update closely so as to preserve the information flow suggested by the dual structure. It is also possible to derive these updates by using a Lagrangian, similarly to the ADMM case. This derivation does not yield the update of $\boldsymbol{\eta}_k$ we used. Nor does it connect to the $t_k$ update used in PVI, which is inspired by similar updates used in expectation-propagation (Minka, 2001). Our modification here is important to connect such updates used in variational methods to those used in ADMM. These details are further discussed in App. C.4. We also remark here that Lagrangian formulations of VB have been used for dual optimization of Gaussian latent models (Khan, 2014; Khan et al., 2013), but they do not exploit the duality of exponential families, instead they use mean-covariance parametrizations of Gaussians.

### 3.4 DERIVING FEDERATED ADMM FROM BAYESIAN DUALITY

We now derive the original ADMM as a special case of Bayesian-ADMM and then propose new extensions of it. All of the derivations that follow rely on a simple two-step procedure:

1. Choose an EF and identify its $\boldsymbol{\lambda}$, $\boldsymbol{\mu}$ and $\mathbf{T}(\boldsymbol{\theta})$, for example, using Nielsen & Garcia (2009).
2. Derive the form of the natural gradient, for example, using Khan & Rue (2023).

To derive ADMM as a special case, we choose the EF to be isotropic Gaussian $q(\boldsymbol{\theta}) = \mathcal{N}(\boldsymbol{\theta} \, | \, \mathbf{m}, \mathbf{I})$ with mean $\mathbf{m}$. For the two steps, we can write the necessary quantities as follows:

$$\boldsymbol{\lambda} = \mathbf{m}, \qquad \boldsymbol{\mu} = \mathbf{m}, \qquad \mathbf{T}(\boldsymbol{\theta}) = \boldsymbol{\theta}, \qquad \nabla \mathcal{L}_k(\mathbf{m}) = \mathbb{E}_{q_k}[\nabla \ell_k]. \tag{21}$$

The last equality is due to Bonnet's theorem (Rezende et al., 2014). Plugging these equations in Eqs. 15 to 17, we can write a simplified version that closely resembles ADMM. A full derivation is

**Algorithm 1** (IVON-ADMM) Adam-like variant of ADMM implemented using IVON (Shen et al., 2024). Additional steps or modifications when compared to regular federated ADMM are highlighted in red. A derivation is in App. D.3 and details on IVON in App. D.4.

**Hyperparameters:** $\delta > 0, \rho > 0, \gamma > 0$.
**Initialize:** $\mathbf{v}_k \leftarrow 0, \mathbf{u}_k \leftarrow 0, \mathbf{m}_g \leftarrow 0, \mathbf{s}_g \leftarrow \delta, \alpha \leftarrow 1/(1 + \rho K)$.
1: **while** not converged **do**
2:     Broadcast $\mathbf{m}_g$ and $\mathbf{s}_g$ to all clients.
3:     **for** each client $1, \ldots, K$ in parallel **do**
4:        Form the loss $\ell(\boldsymbol{\theta}) \leftarrow \frac{1}{\rho}\left(\ell_k(\boldsymbol{\theta}) + \boldsymbol{\theta}^\top \mathbf{v}_k - \frac{1}{2}\boldsymbol{\theta}^\top \operatorname{diag}(\mathbf{u}_k)\boldsymbol{\theta}\right)$
5:        Get $(\mathbf{m}_k, \mathbf{s}_k)$ by training on $\ell(\boldsymbol{\theta})$ using IVON with prior $\mathcal{N}(\boldsymbol{\theta} \,|\, \mathbf{m}_g, \operatorname{diag}(\mathbf{s}_g)^{-1})$
6:        $\mathbf{v}_k \leftarrow \mathbf{v}_k + \gamma\left(\mathbf{s}_k \mathbf{m}_k - \mathbf{s}_g \mathbf{m}_g\right)$
7:        $\mathbf{u}_k \leftarrow \mathbf{u}_k + \gamma\left(\mathbf{s}_k - \mathbf{s}_g\right)$
8:     **end for**
9:     Gather $\mathbf{m}_k, \mathbf{v}_k$ and $\mathbf{s}_k, \mathbf{u}_k$ from all clients.
10:    $\mathbf{s}_g \leftarrow (1 - \alpha)\operatorname{Mean}(\mathbf{s}_{1:K}) + \alpha\left[\delta + \operatorname{Sum}(\mathbf{u}_{1:K})\right]$
11:    $\mathbf{m}_g \leftarrow \left[(1 - \alpha)\operatorname{Mean}(\mathbf{s}_{1:K}\mathbf{m}_{1:K}) + \alpha\operatorname{Sum}(\mathbf{v}_{1:K})\right] / \mathbf{s}_g$
12: **end while**

given in App. D.1, and here we illustrate simplification of the client updates. We start with Eq. 15 where, after renaming $\boldsymbol{\eta}_k$ to $\mathbf{v}_k$, the update simplifies to

$$\mathbf{m}_k \leftarrow \operatorname*{argmin}_{\mathbf{m}_k} \, \mathbb{E}_{\mathcal{N}(\boldsymbol{\theta} \,|\, \mathbf{m}_k, \mathbf{I})}[\ell_k(\boldsymbol{\theta}) + \mathbf{v}_k^\top \boldsymbol{\theta}] + \frac{\rho}{2}\|\mathbf{m}_k - \mathbf{m}_g\|^2. \tag{22}$$

The simplification is obtained by using the definition of the KL divergence for isotropic Gaussians, followed by some rearrangement. The update takes a similar form to ADMM where a linear term $\mathbf{v}_k^\top \boldsymbol{\theta}$ is added to the loss. The main difference is that there is an expectation over the isotropic Gaussian distribution, which is similar to those used in sharpness-aware minimization (Foret et al., 2021; Möllenhoff & Khan, 2023). Similarly, by using the definitions in Eq. 21, we can simplify Eq. 16 to $\mathbf{v}_k \leftarrow \mathbf{v}_k + \rho(\mathbf{m}_k - \mathbf{m}_g)$. This update has a similar property to Eq. 8 where, after every client update, the dual $\mathbf{v}_k = -\mathbb{E}_{q_k}[\nabla \ell_k]$ is the expected gradient.

To recover ADMM as a special case, we employ the delta method (Khan & Rue, 2023, Sec. 1.3.1) where we set $\mathbb{E}[\ell_k(\boldsymbol{\theta})] \approx \ell_k(\mathbf{m}_k)$. It is easy to see that, after this approximation, the update is identical to Eq. 5 if we rename $\mathbf{m}_k$ and $\mathbf{m}_g$ to $\boldsymbol{\theta}_k$ and $\boldsymbol{\theta}_g$, respectively. The simplification of other lines follows in a similar manner and is detailed in App. D.1. We note that the method of Swaroop et al. (2025) does not recover ADMM mainly because it lacks the last KL term in Eq. 20.

### 3.5 New Newton-like and Adam-like Variants of Federated ADMM

We now present two new extensions of federated ADMM. The extensions are obtained by specializing Bayesian-ADMM to Gaussians with full covariance and diagonal covariance respectively. We first describe the Newton-like extension which uses full-covariance Gaussians for both server $q_g(\boldsymbol{\theta}) = \mathcal{N}(\boldsymbol{\theta} \,|\, \mathbf{m}_g, \mathbf{S}_g^{-1})$ and clients $q_k(\boldsymbol{\theta}) = \mathcal{N}(\boldsymbol{\theta} \,|\, \mathbf{m}_k, \mathbf{S}_k^{-1})$. The derivation exactly follows the two-step procedure described earlier and is detailed in App. D.2. Below, we give a brief summary.

When using full-Gaussian, the Bayesian-ADMM client step in Eq. 15 essentially adds an additional quadratic term (highlighted in red),

$$(\mathbf{m}_k, \mathbf{S}_k) \leftarrow \operatorname*{argmin}_{\mathbf{m}_k, \mathbf{S}_k} \mathbb{E}_{q_k}[\ell_k(\boldsymbol{\theta}) + \mathbf{v}_k^\top \boldsymbol{\theta} - \tfrac{1}{2}\boldsymbol{\theta}^\top \mathbf{V}_k \boldsymbol{\theta}] + \rho \operatorname{KL}(q_k \,\|\, q_g). \tag{23}$$

This happens because the sufficient statistics is now $\mathbf{T}(\boldsymbol{\theta}) = (\boldsymbol{\theta}, \boldsymbol{\theta}\boldsymbol{\theta}^\top)$, giving rise to an additional dual variable, denoted by the matrix $\mathbf{V}_k$. As shown in App. D.2, this dual variable is related to the Hessian, that is, $\mathbf{V}_k = \mathbb{E}_{q_k}[\nabla^2 \ell_k]$ after every local update. Due to this, the local and global updates can be implemented in the form of a Newton step. Similarly to Newton's method, the new method converges in a single-step on quadratic objectives, which we show in App. D.2 and also later verify in the experiments section.

We also derive a scalable Adam-like variant of ADMM by using Gaussians with diagonal covariances. The resulting method is shown in Alg. 1 and makes use of the Improved Variational Online Newton

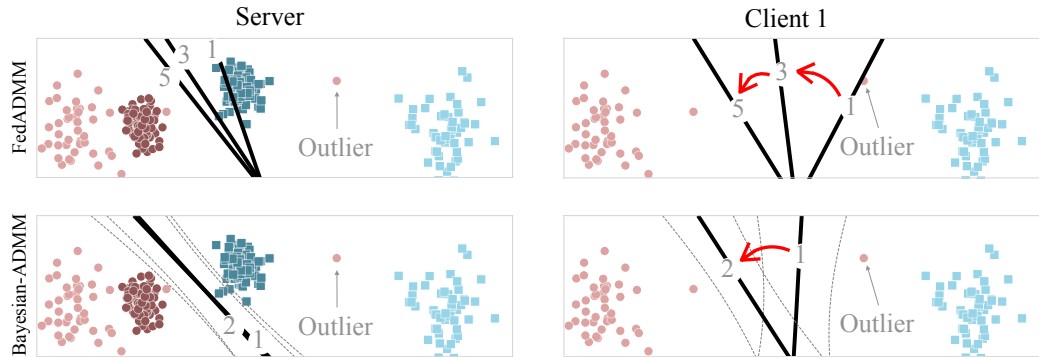

Figure 4: A single outlier that slows down ADMM (top row) poses no issues for our new Bayesian version Bayesian-ADMM (bottom row). The server (left column) with ADMM takes 5 iterations while with Bayesian-ADMM it only needs 2 iterations (decision boundaries are numbered with iteration number). Client 1 (right column) is the source of the issue which takes 5 iterations to ignore the outlier, while with Bayes, it is much faster due to the use of uncertainty (gray lines). Light points indicate data on client 1, dark points indicate data on client 2 (not shown).

(IVON) optimizer of Shen et al. (2024) to solve the client subproblem. For this reason, we call it IVON-ADMM. A detailed derivation of this method is given in App. D.3. Implementation of IVON-ADMM is nearly identical to the standard ADMM and costs are nearly identical too. Lines 4 and 5 in Alg. 1 solve the client subproblem in Bayesian-ADMM which takes a similar form to Eq. 23 but with diagonal covariances. In practice, this is a standard application of the IVON optimizer, and more details on how exactly it is used are in App. D.4. Lines 6 and 7 implement the dual update in Eq. 16 in Bayesian-ADMM and involve just simple pointwise arithmetic operations. We have introduced a different step-size $\gamma$ in the dual, as we found this to improve the practical performance. Finally, lines 9-11 implement the server step in Eq. 17 in Bayesian-ADMM.

All changes are simple pointwise operations and easy to implement, and the IVON optimizer has no overhead over training with Adam (Shen et al., 2024). Therefore the complexity is the same as federated ADMM algorithms such as FedDyn (Acar et al., 2021). Communication cost is doubled compared to ADMM as we send both a mean and diagonal variance vector, as in other preconditioned or Bayesian methods (Swaroop et al., 2025). This is expected to help in cases with heterogeneity.

## 4 NUMERICAL EXPERIMENTS

### 4.1 ILLUSTRATIVE EXAMPLES

In Fig. 4 we show that the Newton-like variant of Bayesian-ADMM can better handle heterogeneous and noisy data. We compare to ADMM on a classification task: data is split over two clients, and client 1 contains an outlier. When using a linear classifier, federated ADMM takes a total of five communication rounds to converge, primarily due to the outlier. In contrast, Bayesian-ADMM quickly converges in a single round. This is achieved by assigning a higher posterior uncertainty to the outlier, which enables the algorithm to discount it and converge faster. Details are in App. E.1.

In Fig. 5a, we show a linear regression problem with full Gaussian posteriors (details in App. E.2), where Bayesian-ADMM with full covariances converges in a single round. This is similar to Newton methods on quadratic functions. There are no previous generalizations of ADMM with such a property. In Fig. 5b we compare the Newton-like Bayesian-ADMM method with full covariances to PVI and BregmanADMM on logistic regression (details are in App. E.3). We find that PVI without damping diverges, whereas PVI with damping converges but slower than Bayesian-ADMM.

### 4.2 EVALUATION ON FEDERATED DEEP LEARNING BENCHMARKS

We follow a similar setup to Swaroop et al. (2025) and train neural nets on image datasets. We consider different levels of heterogeneity on the clients and different numbers of clients. We focus on comparing IVON-ADMM (Alg. 1) to (i) the distributed Bayesian algorithms FedLap and FedLap-Cov

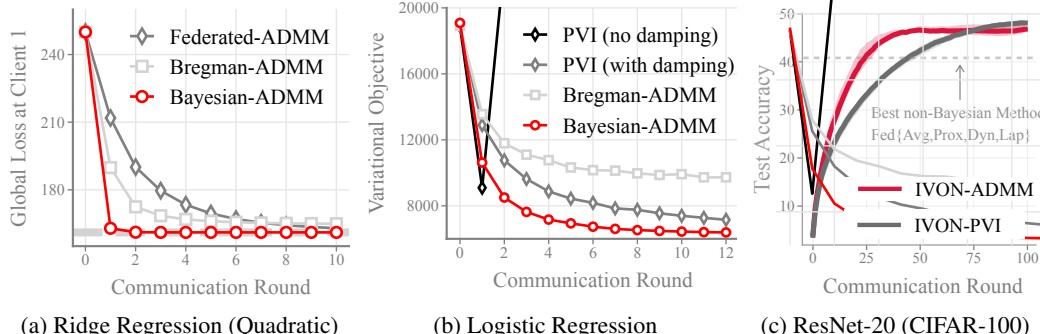

(a) Ridge Regression (Quadratic)  (b) Logistic Regression  (c) ResNet-20 (CIFAR-100)

Figure 5: (a) Bayesian-ADMM converges in a single round for quadratics, whereas other methods need many steps. (b) PVI can diverge on logistic regression (MNIST). Damping used by Ashman et al. (2022) improves this, but is slower than Bayesian-ADMM, also for CIFAR-100 in plot (c).

Table 1: IVON-ADMM outperforms all baselines in terms of accuracy and NLL. Averaging over the posterior in IVON-ADMM (as opposed to IVON-ADMM@$\mathbf{m}_g$) often improves performance.

| Scenario | Method | Test accuracy (↑ larger is better) | | | Test NLL (↓ smaller is better) | | |
|---|---|---|---|---|---|---|---|
| | | 10 rounds | 25 rounds | 50 rounds | 10 rounds | 25 rounds | 50 rounds |
| MLP, 100 clients heterog. FMNIST | FedAvg/FedProx | 73.9±0.3 | 78.7±0.3 | 81.8±0.3 | 0.73±0.01 | 0.60±0.01 | 0.53±0.01 |
| | FedDyn | 75.7±0.4 | 81.4±0.2 | 82.2±0.3 | 0.66±0.01 | 0.53±0.02 | 0.51±0.01 |
| | FedLap | 73.9±0.3 | 79.4±0.2 | 82.4±0.3 | 0.68±0.01 | 0.59±0.01 | 0.53±0.01 |
| | FedLap-Cov | 76.9±0.7 | 81.3±0.2 | **83.0±0.1** | 0.64±0.01 | 0.54±0.00 | 0.49±0.01 |
| | IVON-ADMM@$\mathbf{m}_g$ | **78.7±0.1** | 82.2±0.1 | 82.9±0.2 | **0.62±0.00** | **0.50±0.00** | **0.48±0.00** |
| | IVON-ADMM | **78.7±0.1** | 82.3±0.1 | 83.0±0.2 | **0.62±0.00** | **0.50±0.00** | **0.48±0.00** |
| MLP, 10 clients homogeneous FMNIST[(*)] | FedAvg | 72.3±0.4 | 77.7±0.3 | 80.0±0.2 | 0.70±0.00 | 0.61±0.01 | 0.56±0.01 |
| | FedProx | 72.2±0.3 | 77.4±0.1 | 80.3±0.1 | 0.71±0.00 | 0.61±0.01 | 0.56±0.01 |
| | FedDyn | 75.3±0.8 | 77.5±0.8 | 78.2±0.5 | 0.67±0.01 | 0.63±0.02 | 0.63±0.02 |
| | FedLap | 72.1±0.2 | 77.1±0.1 | 80.2±0.1 | 0.71±0.01 | 0.62±0.01 | 0.57±0.01 |
| | FedLap-Cov | 75.0±0.6 | 79.8±0.4 | 81.8±0.1 | 0.67±0.01 | 0.59±0.01 | 0.56±0.01 |
| | IVON-ADMM@$\mathbf{m}_g$ | 80.4±0.2 | 83.1±0.1 | 83.4±0.1 | **0.56±0.00** | **0.53±0.00** | 0.60±0.01 |
| | IVON-ADMM | **80.6±0.2** | **83.5±0.1** | **84.1±0.2** | **0.55±0.00** | **0.50±0.00** | **0.53±0.01** |
| CNN (Zenke et al., 2017) 10 clients heterog. CIFAR-10 | FedAvg/FedProx | 73.8±1.5 | 75.0±0.9 | 75.4±0.8 | 0.9±0.0 | 1.2±0.1 | 1.5±0.1 |
| | FedDyn | 72.7±0.9 | 77.4±0.6 | 79.4±0.4 | 1.0±0.0 | 0.8±0.0 | 0.7±0.0 |
| | FedLap | **74.8±1.3** | 78.0±1.2 | 79.5±1.4 | **0.7±0.0** | **0.6±0.0** | **0.6±0.0** |
| | FedLap-Cov | **75.1±1.1** | 77.6±0.7 | 79.2±0.9 | **0.7±0.0** | 0.7±0.0 | **0.6±0.0** |
| | IVON-ADMM@$\mathbf{m}_g$ | 71.5±1.8 | 78.9±0.8 | **80.3±0.6** | 0.8±0.0 | **0.6±0.0** | **0.6±0.0** |
| | IVON-ADMM | 71.5±1.8 | 79.0±0.8 | **80.3±0.6** | 0.8±0.0 | **0.6±0.0** | **0.6±0.0** |
| CNN (Acar et al., 2021) 10 clients heterog. CIFAR-10 | FedAvg/FedProx | **64.3±2.0** | 65.9±1.6 | 66.3±1.4 | 1.7±0.1 | 2.2±0.1 | 2.6±0.1 |
| | FedDyn | 63.6±1.1 | 64.7±0.6 | 65.4±1.0 | 2.0±0.6 | 3.6±1.3 | 2.6±0.1 |
| | FedLap | 60.2±2.4 | 66.4±1.1 | 66.5±1.2 | 1.1±0.0 | 1.3±0.1 | 1.8±0.1 |
| | FedLap-Cov | 58.4±2.4 | 65.4±1.1 | 67.5±1.2 | 1.2±0.0 | **1.0±0.0** | **1.0±0.0** |
| | IVON-ADMM@$\mathbf{m}_g$ | **63.8±1.4** | **69.5±0.8** | 70.2±0.7 | **1.0±0.0** | **1.0±0.0** | 1.5±0.0 |
| | IVON-ADMM | **63.8±1.4** | **69.5±0.8** | **70.3±0.6** | **1.0±0.0** | **1.0±0.0** | 1.4±0.0 |

(Swaroop et al., 2025), (ii) FedDyn (Acar et al., 2021) which is the best-performing federated ADMM-style algorithm, and (iii) standard baseline algorithms for federated learning, FedAvg (McMahan et al., 2016) and FedProx (Li et al., 2020). Further hyperparameters and details are in App. F.

Overall, we find that IVON-ADMM outperforms all baselines in most cases; IVON-ADMM has lower test loss than baselines, showing the benefits of a Bayesian method; It is also significantly cheaper than the second-best method FedLap-Cov. Ensembling predictions over the posterior at test-time usually improves the performance. In what follows, we describe the experiments in detail.

**Experimental setup.** Following Swaroop et al. (2025), we consider fully connected neural networks on the MNIST and FashionMNIST datasets, and convolutional networks on CIFAR-10 and 100, We use homogeneous and heterogeneous splits on $K = 10$ and $K = 100$ clients. On CIFAR-100 we use a ResNet-20 architecture. For all heterogeneous splits, we follow previous work and sample Dirichlet distributions that decide how many points per class go into each client (Swaroop et al., 2025). For the highly-heterogeneous MNIST split ($K = 100$ clients), we assign data from only two random classes per client (McMahan et al., 2016).

Table 2: Test accuracy and NLL after 25, 50 and 100 rounds, with mean and standard deviations over 3 runs. We see that IVON-ADMM outperforms all baselines in both scenarios.

| Scenario | Method | Test accuracy (↑ larger is better) | | | Test NLL (↓ smaller is better) | | |
|---|---|---|---|---|---|---|---|
| | | 25 rounds | 50 rounds | 100 rounds | 25 rounds | 50 rounds | 100 rounds |
| MLP, 100 clients highly heterog. MNIST | FedAvg/FedProx | 89.6±0.4 | 91.8±0.5 | 94.0±0.4 | 0.59±0.01 | 0.36±0.01 | 0.22±0.01 |
| | FedDyn | 89.8±0.1 | 92.6±0.2 | 94.9±0.1 | 0.60±0.01 | 0.33±0.01 | 0.18±0.00 |
| | FedLap | 89.7±0.3 | 92.0±0.4 | 94.4±0.3 | 0.63±0.01 | 0.38±0.01 | 0.22±0.01 |
| | FedLap-Cov | **91.0**±0.4 | 92.9±0.4 | 94.9±0.3 | 0.47±0.01 | 0.29±0.01 | 0.18±0.01 |
| | IVON-ADMM@$\mathbf{m}_g$ | **91.0**±0.1 | **94.2**±0.1 | **96.0**±0.1 | **0.31**±0.02 | **0.19**±0.00 | **0.13**±0.00 |
| | IVON-ADMM | **91.0**±0.1 | **94.2**±0.1 | **96.0**±0.1 | **0.31**±0.03 | **0.19**±0.01 | **0.13**±0.00 |
| ResNet-20, 10 clients, heterog. CIFAR-100 | FedAvg/FedProx | 37.9±0.4 | 40.4±0.9 | 39.8±0.8 | 2.4±0.0 | 2.6±0.1 | 3.4±0.1 |
| | FedDyn | 38.4±0.5 | 39.2±0.8 | 39.6±0.4 | 3.5±0.3 | 3.2±0.2 | 3.2±0.1 |
| | FedLap | **40.1**±1.1 | 40.4±1.0 | 39.7±0.7 | 2.9±0.1 | 3.0±0.0 | 3.2±0.0 |
| | IVON-ADMM@$\mathbf{m}_g$ | **40.0**±1.0 | **46.2**±0.4 | **46.5**±0.6 | **2.3**±0.0 | **2.1**±0.0 | **2.3**±0.1 |
| | IVON-ADMM | **40.0**±1.0 | **46.2**±0.4 | **46.6**±0.6 | **2.3**±0.0 | **2.1**±0.0 | **2.2**±0.1 |

Our results are summarized in Tables 1, 2 and 4, as well as Fig. 2. In these tables, the IVON-ADMM@$\mathbf{m}_g$ method predicts at the server's posterior mean $\mathbf{m}_g$, while the IVON-ADMM method ensembles predictions of 32 samples from $q_g$. We average accuracy and test loss (NLL) over three seeds after 10, 25, 50 and 100 rounds.

**IVON-ADMM is overall the best method.** We compare IVON-ADMM to baselines for 10 and 100-client splits on FMNIST and CIFAR-10 (Table 1). It is much better than all five baselines on FMNIST in terms of test NLL and accuracy, at all rounds. In particular, we (i) improve upon FedDyn (which is Federated ADMM), showing the importance of including posterior covariances in IVON-ADMM; (ii) improve upon FedLap and FedLap-Cov, showing the importance of the Bayesian-ADMM update equations and of using IVON instead of a covariance obtained via Laplace approximations.

**For longer runs, IVON-ADMM significantly outperforms all other methods.** We compare IVON-ADMM to baselines for a 100-client highly heterogeneous MNIST split, and a 10-client heterogeneous CIFAR-100 split, in Table 2. We see that IVON-ADMM significantly outperforms all baselines, showing the benefits with more clients and over more communication rounds. After 100 rounds on CIFAR-100, accuracy is improved by 6.7% and test NLL by 0.9.

**IVON-ADMM has less overhead than FedLap-Cov.** IVON-ADMM has similar cost and runtime to FedAvg, FedProx, FedDyn and FedLap, despite estimating a (diagonal) covariance matrix. FedLap-Cov requires a Laplace approximation to the covariance matrix, which is slow and memory-intensive: FedLap-Cov is 4 times slower than FedLap on MNIST, 7 times slower on CIFAR-10, and prohibitively slow on CIFAR-100, even though we use LaplaceRedux (Daxberger et al., 2021).

**Faster convergence than IVON-PVI.** We introduce a new method, IVON-PVI, which is a special case of our IVON-ADMM (with $\alpha = 1$, $\rho = 1$). In Fig. 5c we see that the additional step-sizes in IVON-ADMM means that it converges quicker than IVON-PVI, though the final solution is of comparable quality. Additional results are in App. G.1.

**Sensitivity to Hyperparameters.** We perform an ablation study over the step size ($\rho$) and the temperature ($\tau$) of IVON-ADMM. The results are discussed in App. G.2, where the main findings are: (1) A too-large $\rho$ leads to lower accuracy, but too small $\rho$ can cause the method to diverge. (2) For the considered dataset and architecture, the optimal temperature is at around $0.1 - 0.2$ and deviating too much from it causes degradations in performance.

## 5 DISCUSSION

We introduced a Bayesian duality, from which we naturally extended ADMM to optimize over distributions. For Gaussians with fixed variance, we recover regular ADMM, and general Gaussians give Newton-like methods and IVON-ADMM. These show good performance when compared to recent baselines. Other approximating distributions may lead to new interesting splitting algorithms, and more generally, our work opens up new research paths to extend and improve primal-dual algorithms using Bayesian ideas. Extending the framework to decentralized, asynchronous, and adaptive settings, as well as variants of IVON-ADMM with non-diagonal covariances are promising directions for future work. Bayesian-ADMM defines a new class of optimization algorithms, and obtaining a deeper understanding of its fundamental nature remains an open problem.

ACKNOWLEDGEMENTS

This work is supported by JST CREST Grant Number JP-MJCR2112. This work used computational resources on the TSUBAME4.0 supercomputer provided by Institute of Science Tokyo. This material is based upon work supported by the National Science Foundation under Grant No. IIS-2107391. Any opinions, findings, and conclusions or recommendations expressed in this material are those of the author(s) and do not necessarily reflect the views of the National Science Foundation.

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

# A  ADMM DERIVATION FOR EQS. 2 AND 4 TO 8

To derive the optimality condition given in Eq. 2, we note that $\ell_0(\boldsymbol{\theta}_g) = \frac{1}{2}\|\boldsymbol{\theta}_g\|^2$. Therefore, taking the derivative of the objective with respect to $\theta$ gives us

$$\sum_{k=1}^{K} \nabla\ell_k(\boldsymbol{\theta}_g^*) + \boldsymbol{\theta}_g^* = 0 \quad\Longrightarrow\quad \boldsymbol{\theta}_g^* = -\sum_{k=1}^{K}\nabla\ell_k(\boldsymbol{\theta}_g^*).$$

A Lagrangian can be formed as follows:

$$\min_{\boldsymbol{\theta}_g, \boldsymbol{\theta}_{1:K}} \max_{\mathbf{v}_{1:K}} \sum_{k=1}^{K}\left[\ell_k(\boldsymbol{\theta}_k) + \mathbf{v}_k^\top(\boldsymbol{\theta}_k - \boldsymbol{\theta}_g)\right] + \ell_0(\boldsymbol{\theta}_g). \tag{24}$$

For the Lagrangian, the derivative with respect to $\boldsymbol{\theta}_k$ yields

$$\nabla[\ell_k(\boldsymbol{\theta}_k^*) + (\mathbf{v}_k^*)^\top\boldsymbol{\theta}_k^*] = 0 \quad\Longrightarrow\quad \mathbf{v}_k^* = -\nabla\ell_k(\boldsymbol{\theta}_k^*),$$

which is the second condition. Similarly, derivatives with respect to $\boldsymbol{\theta}_g$ gives us

$$\nabla\left[\ell_0(\boldsymbol{\theta}_g^*) - \sum_{k=1}^{K}(\mathbf{v}_k^*)^\top\boldsymbol{\theta}_g^*\right] = 0 \quad\Longrightarrow\quad \boldsymbol{\theta}_g^* = \sum_{k=1}^{K}\mathbf{v}_k^*.$$

Then defining the sum of $\mathbf{v}_k^*$ as $\mathbf{v}_g^*$, we get the third and the fourth condition. Finally, taking the derivative with respect to $\mathbf{v}_k$, we get the first condition $\boldsymbol{\theta}_g^* = \boldsymbol{\theta}_k^*$.

We now provide a short derivation of federated ADMM as shown in Eqs. 5 to 7. The method is a standard application of ADMM to the consensus problems with regularization, see Boyd et al. (2011, Section 7.1.1). First, we collect from Eq. 24 all the terms depending on $\boldsymbol{\theta}_k$, and add a quadratic proximity term to optimize the objective as follows,

$$\boldsymbol{\theta}_k \leftarrow \arg\min_{\boldsymbol{\theta}_k} \ell_k(\boldsymbol{\theta}_k) + \mathbf{v}_k^\top\boldsymbol{\theta}_k + \frac{\rho}{2}\|\boldsymbol{\theta}_k - \boldsymbol{\theta}_g\|^2, \tag{25}$$

with $\rho > 0$ as the (inverse) step-size. We locally optimize this to update $\boldsymbol{\theta}_k$, which is shown in Eq. 5. The update of $\mathbf{v}_k$ follows in a similar fashion,

$$\mathbf{v}_k \leftarrow \arg\max_{\mathbf{v}_k} \mathbf{v}_k^\top(\boldsymbol{\theta}_g - \boldsymbol{\theta}_k) - \frac{1}{2\rho}\|\mathbf{v}_k - \mathbf{v}_k^{\text{old}}\|^2 \quad\Longrightarrow\quad \mathbf{v}_k \leftarrow \mathbf{v}_k^{\text{old}} + \rho(\boldsymbol{\theta}_k - \boldsymbol{\theta}_g). \tag{26}$$

This is the update given in Eq. 6. The third line of the algorithm similarly optimizes,

$$\boldsymbol{\theta}_g \leftarrow \arg\min_{\boldsymbol{\theta}_g} \ell_0(\boldsymbol{\theta}_g) - \sum_{k=1}^{K}\mathbf{v}_k^\top\boldsymbol{\theta}_g + \frac{\rho}{2}\sum_{k=1}^{K}\|\boldsymbol{\theta}_k - \boldsymbol{\theta}_g\|^2.$$

For the special case when $\ell_0(\boldsymbol{\theta}_g) = \frac{1}{2}\|\boldsymbol{\theta}_g\|^2$, the server update only requires an average over $\boldsymbol{\theta}_k$ and a sum over $\mathbf{v}_k$, as shown below by taking the fixed point of the above update:

$$\boldsymbol{\theta}_g - \sum_{k=1}^{K}\mathbf{v}_k - \rho\sum_{k=1}^{K}\boldsymbol{\theta}_k + \rho K\boldsymbol{\theta}_g = 0$$

$$\Leftrightarrow \quad \boldsymbol{\theta}_g \leftarrow \alpha\sum_{k=1}^{K}\mathbf{v}_k + (1-\alpha)\frac{1}{K}\sum_{k=1}^{K}\boldsymbol{\theta}_k, \tag{27}$$

where we define $\alpha = 1/(\rho K + 1)$.

Finally, to see why the dual variables correspond to gradients as in Eq. 8, we consider the optimality condition of the client update in Eq. 25:

$$\nabla\ell_k(\boldsymbol{\theta}_k) + \mathbf{v}_k^{\text{old}} + \rho(\boldsymbol{\theta}_k - \boldsymbol{\theta}_g) = 0 \quad\Leftrightarrow\quad \mathbf{v}_k^{\text{old}} + \rho(\boldsymbol{\theta}_k - \boldsymbol{\theta}_g) = -\nabla\ell_k(\boldsymbol{\theta}_k). \tag{28}$$

Plugging this into the dual update on the right in Eq. 26, we immediately get $\mathbf{v}_k = -\nabla\ell_k(\boldsymbol{\theta}_k)$.

Table 3: A summary of exponential-families used in the paper showing natural parameters $\boldsymbol{\lambda}$, sufficient statistics $\mathbf{T}(\boldsymbol{\theta})$, and expectation parameters $\boldsymbol{\mu}$, reproduced from Khan & Rue (2023, Table 2).

| Distribution | $\boldsymbol{\lambda}$ | $\mathbf{T}(\boldsymbol{\theta})$ | $\boldsymbol{\mu}$ | Resulting Method |
|---|---|---|---|---|
| $\mathcal{N}(\boldsymbol{\theta} \mid \mathbf{m}, \mathbf{I})$ | $\mathbf{m}$ | $\boldsymbol{\theta}$ | $\mathbf{m}$ | ADMM |
| $\mathcal{N}(\boldsymbol{\theta} \mid \mathbf{m}, \mathbf{S}^{-1})$, fixed $\mathbf{S}$ | $\mathbf{Sm}$ | $\boldsymbol{\theta}$ | $\mathbf{m}$ | Preconditioned ADMM |
| $\mathcal{N}(\boldsymbol{\theta} \mid \mathbf{m}, \mathbf{S}^{-1}), \mathbf{S} = \mathrm{diag}(\mathbf{s})$ | $\begin{pmatrix} \mathbf{sm} \\ -\frac{1}{2}\mathbf{s} \end{pmatrix}$ | $\begin{pmatrix} \boldsymbol{\theta} \\ \boldsymbol{\theta}^2 \end{pmatrix}$ | $\begin{pmatrix} \mathbf{m} \\ \mathbf{m}^2 + 1/\mathbf{s} \end{pmatrix}$ | Alg. 1 |
| $\mathcal{N}(\boldsymbol{\theta} \mid \mathbf{m}, \mathbf{S}^{-1})$ | $\begin{pmatrix} \mathbf{Sm} \\ -\frac{1}{2}\mathbf{S} \end{pmatrix}$ | $\begin{pmatrix} \boldsymbol{\theta} \\ \boldsymbol{\theta}\boldsymbol{\theta}^\top \end{pmatrix}$ | $\begin{pmatrix} \mathbf{m} \\ \mathbf{mm}^\top + \mathbf{S}^{-1} \end{pmatrix}$ | App. D.2 |

## B  BACKGROUND ON VARIATIONAL BAYES

### B.1  EXPONENTIAL FAMILIES

In this appendix, we provide a short self-contained introduction to exponential families. Many commonly used distributions such as Gaussian, categorical or gamma distributions belong to an exponential family. Viewing them as such provides us with a rich dual structure which we exploit in this paper. Table 3 provides an overview over all exponential families used in this paper.

Here, we give a short introduction and refer the interested reader to Wainwright et al. (2008, Sec. 3) for more details and a rigorous treatment. Similar to Jaynes (1957), we introduce exponential families as maximum-entropy solutions of a constrained optimization problem. We assume a base measure which has density $h(\boldsymbol{\theta})$ with respect to the Lebesgue measure and then search for a density $q(\boldsymbol{\theta})$ which has maximum entropy (relative to $h$) and whose expectation under sufficient statistics $\mathbf{T}(\boldsymbol{\theta})$ matches given expectations or moments $\boldsymbol{\mu}$. There are many distributions which share the same moments $\boldsymbol{\mu}$, but only one has the maximum entropy, see Fig. 6a for an example with uniform density $h(\boldsymbol{\theta}) = 1$. The maximum entropy density is given as the solution to the following problem:

$$q_{\boldsymbol{\mu}} = \operatorname{argmin}_q \; \mathrm{KL}(q \,\|\, h) \; \text{ s.t. } \; \mathbb{E}_q[\mathbf{T}(\boldsymbol{\theta})] = \boldsymbol{\mu}, \; q(\boldsymbol{\theta}) \geq 0, \; \int q(\boldsymbol{\theta})\mathrm{d}\boldsymbol{\theta} = 1. \tag{29}$$

Similar to Jaynes (1957, Sec. 2), introducing Lagrange multipliers $\boldsymbol{\lambda}$ for the moment constraints and another Lagrange multiplier $\alpha$ for the normalization constraint, we get the Lagrangian,

$$\mathcal{L}(q, \boldsymbol{\lambda}, \alpha) = \mathrm{KL}(q \,\|\, h) + \langle \boldsymbol{\lambda}, \mathbb{E}_q[\mathbf{T}(\boldsymbol{\theta})] - \boldsymbol{\mu} \rangle + \alpha \left( 1 - \int q(\boldsymbol{\theta})\mathrm{d}\boldsymbol{\theta} \right). \tag{30}$$

Taking the first variation of the Lagrangian with respect to $q$ and setting it to zero, we arrive at,

$$q_{\boldsymbol{\mu}}(\boldsymbol{\theta}) = h(\boldsymbol{\theta}) \exp(\langle \boldsymbol{\lambda}, \mathbf{T}(\boldsymbol{\theta}) \rangle - \alpha), \tag{31}$$

which for $\alpha = A(\boldsymbol{\lambda}) = \log \int \exp(\langle \boldsymbol{\lambda}, \mathbf{T}(\boldsymbol{\theta}) \rangle)h(\boldsymbol{\theta})\mathrm{d}\boldsymbol{\theta}$ is the definition of the exponential family as provided in the main text in Eq. 10. We also see that the Lagrange multipliers $\boldsymbol{\lambda}$ correspond to the natural parameters of the exponential family.

The Lagrangian we used in the main text to derive our Bayesian-ADMM method closely follows this dual structure of the exponential family. The above derivation provides an additional motivation why one should use the natural parameters $\boldsymbol{\lambda}$ in the dual update of our method in Eq. 16. They live in the correct space of Lagrange multipliers to constraints on the expectation parameters.

As mentioned in the main text, Lagrange multipliers $\boldsymbol{\lambda}$ and moments $\boldsymbol{\mu}$ form a dual coordinate system, where $\boldsymbol{\mu} = \nabla A(\boldsymbol{\lambda}) = \mathbb{E}_q[\mathbf{T}(\boldsymbol{\theta})]$. The map $\nabla A : \Omega \to \mathcal{M}$ maps from the space of valid natural parameters $\Omega = \{\boldsymbol{\lambda} : A(\boldsymbol{\lambda}) < \infty\}$ to the space of valid moments $\mathcal{M} = \{\mathbb{E}_q[\mathbf{T}(\boldsymbol{\theta})] : q \in \mathcal{Q}\}$. The inverse is given by $\nabla A^* : \mathcal{M} \to \Omega$, where $A^*$ denotes the convex conjugate of $A$. For an illustration of the two spaces $\Omega$ and $\mathcal{M}$ on the example of Gaussian distributions, see Fig. 6b.

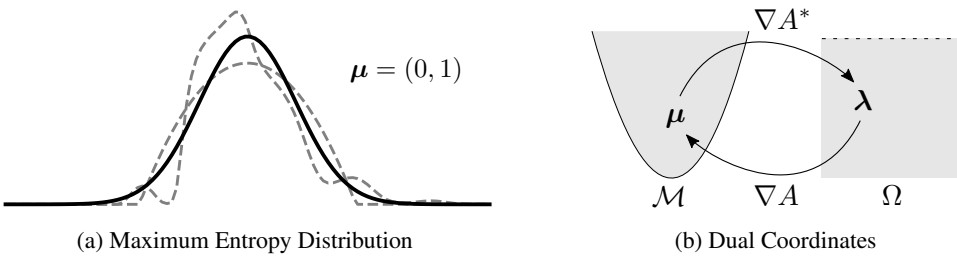

Figure 6: (a) There are many distributions (two possible ones shown in dashed gray lines) which share the same moments $\boldsymbol{\mu}$. Distributions which have maximum entropy among all possible candidates belong to an exponential family (here: the Gaussian shown in solid black). (b) There is a one-to-one correspondence between the moments $\boldsymbol{\mu} \in \mathcal{M}$ and the natural parameters $\boldsymbol{\lambda} \in \Omega$, which are Lagrange multipliers to the moment constraints in the maximum entropy variational problem.

### B.2 DERIVATION OF THE FIXED-POINT EQUATION EQ. 12

We start by writing the natural gradient of the KL divergence as follows,

$$\nabla_{\boldsymbol{\mu}}\mathrm{KL}(q \,\|\, \pi_0) = \nabla_{\boldsymbol{\mu}}\mathbb{E}_q[\log q - \log \pi_0] = \boldsymbol{\lambda} - \nabla_{\boldsymbol{\mu}}\mathbb{E}_q[\ell_0].$$

The last equality follows from Khan & Rue (2023, Eq. 64) and also noting that $\pi_0 \propto \exp(-\ell_0)$. Writing the second term in terms of $\mathcal{L}_0$, we can use the above to write the gradient of Eq. 9 as

$$\sum_{k=0}^{K} \nabla\mathcal{L}_k(\boldsymbol{\mu}) + \boldsymbol{\lambda}.$$

Setting this to 0 at $\boldsymbol{\mu}_g^*$, we get the first equation in Eq. 12. Then, substituting the first equation in Eq. 10 and simplifying, we get the second equation.

## C DERIVATIONS AND DETAILS ON BAYESIAN-ADMM

### C.1 DUAL VARIABLES ARE NATURAL GRADIENTS

The result follows from the optimality condition of the client step in Eq. 15,

$$\nabla_{\boldsymbol{\mu}}\mathcal{L}_k(\boldsymbol{\mu}_k) + \boldsymbol{\eta}_k + \rho(\boldsymbol{\lambda}_k - \boldsymbol{\lambda}_g) = 0, \tag{32}$$

which is directly obtained by taking the derivative in $\boldsymbol{\mu}$ and setting it to zero. The above uses the fact that $\nabla_{\boldsymbol{\mu}}\mathrm{KL}(q_k \,\|\, q_g) = \boldsymbol{\lambda}_k - \boldsymbol{\lambda}_g$, see Khan & Rue (2023, Eqs. 21, 23). Inserting this expression into the dual update in Eq. 16, we get,

$$\boldsymbol{\eta}_k^{\mathrm{new}} \leftarrow \boldsymbol{\eta}_k + \rho(\boldsymbol{\lambda}_k - \boldsymbol{\lambda}_g) = -\nabla_{\boldsymbol{\mu}}\mathcal{L}_k(\boldsymbol{\mu}_k). \tag{33}$$

Thus, the dual variables in Bayesian-ADMM correspond to the natural gradients of the client losses.

### C.2 RELATIONSHIP OF BAYESIAN-ADMM TO METHODS FROM LITERATURE

Here, we discuss connections of Bayesian-ADMM to various methods from the literature.

**Bregman ADMM and augmented Lagrangian methods.** Many variants of ADMM have been proposed since its inception. However, we believe that the dual update in Bayesian-ADMM in Eq. 16,

$$\boldsymbol{\eta}_k \leftarrow \boldsymbol{\eta}_k + \rho(\boldsymbol{\lambda}_k - \boldsymbol{\lambda}_g),$$

is a fundamental departure from other generalizations of ADMM used in the optimization literature (Babagholami-Mohamadabadi et al., 2015; Nodozi & Halder, 2023; Wang et al., 2014; Wang & Banerjee, 2014), which use the primal variables in the dual update. In our setting, this would amount to using $\boldsymbol{\mu}_k - \boldsymbol{\mu}_g$ in the dual update, which is unnatural from a Bayesian viewpoint. Arithmetic in $\boldsymbol{\lambda}$-coordinates corresponds to multiplication and division of distributions, but no such simple interpretation exists in $\boldsymbol{\mu}$-coordinates. Moreover, as we have seen in the previous section of the appendix, the

choice of natural parameters in the dual update ensures the natural condition $\boldsymbol{\eta}_k = -\nabla\mathcal{L}_k(\boldsymbol{\mu}_k)$ at every step of the algorithm, which is essential to our proposal. Nonlinear augmented Lagrangian and Bregman Douglas-Rachford splitting approaches (Bertsekas, 2014; Eckstein & Ferris, 1999; Luque, 1986; Ma et al., 2025; Oikonomidis et al., 2023) come with modified but different dual updates.

**Bayesian Learning Rule.** Bayesian-ADMM is also closely related to the Bayesian learning rule (BLR) (Khan & Rue, 2023), a natural-gradient descent algorithm for VB optimization in the non-federated case. We can show that Bayesian-ADMM mimics the BLR steps when the client updates are iterated until convergence before doing a server update, as stated below.

**Proposition C.1.** *If the client steps 1 and 2 in Eqs. 15 and 16 are iterated until convergence for a given $\boldsymbol{\lambda}_g$, then the server step that follows in Eq. 17 will be equivalent to the Bayesian learning rule (BLR) of Khan & Rue (2023).*

*Proof.* To show this, we note that if we run the client steps iteratively until convergence, then $\boldsymbol{\mu}_k = \boldsymbol{\mu}_g$ and $\boldsymbol{\lambda}_k = \boldsymbol{\lambda}_g$, and the dual vector $\boldsymbol{\eta}_k$ is also equal to the natural gradient at $\boldsymbol{\mu}_g$. Then, we write out the server step of Bayesian-ADMM more explicitly in closed form:

$$\boldsymbol{\lambda}_g = (1-\alpha)\frac{1}{K}\sum_{k=1}^{K}\boldsymbol{\lambda}_k + \alpha\sum_{k=0}^{K}\boldsymbol{\eta}_k. \tag{34}$$

A derivation for this form is given around Eq. 54. The average over $\boldsymbol{\lambda}_k$ is equal to $\boldsymbol{\lambda}_g$, and the update becomes the BLR as written in Khan & Rue (2023, Eq. 6) since $\boldsymbol{\eta}_k$ are the natural gradients. $\square$

### C.3 PARTITIONED VARIATIONAL INFERENCE (PVI)

We first rewrite PVI as introduced in Ashman et al. (2022, Alg. 1), bringing it into the form used by Swaroop et al. (2025). Then, we will discuss differences to Bayesian-ADMM. The first step of Ashman et al. (2022, Alg. 1) is a local update of the distribution $q_k$ which takes the following form,

$$q_k \leftarrow \operatorname{argmax}_{q\in\mathcal{Q}} \int q(\boldsymbol{\theta})\log\frac{q_g(\boldsymbol{\theta})p(\mathbf{y}_k|\boldsymbol{\theta})}{q(\boldsymbol{\theta})t_k(\boldsymbol{\theta})}\mathrm{d}\boldsymbol{\theta} \tag{35}$$

$$= \operatorname{argmin}_{q\in\mathcal{Q}} \mathbb{E}_q[\ell_k(\boldsymbol{\theta}) + \log t_k(\boldsymbol{\theta})] + \mathrm{KL}(q \,\|\, q_g), \tag{36}$$

where for the equality step, we switched from maximization to minimization and replaced the negative log-likelihood with a loss $\ell_k$.

$t_k(\boldsymbol{\theta})$ are site-functions which aim to approximate the likelihood, but they also play a similar rule to Lagrange multipliers in ADMM. Ashman et al. (2022, Alg. 1) then update $t_k$ as follows:

$$t_k(\boldsymbol{\theta}) \leftarrow \frac{q_k(\boldsymbol{\theta})}{q_g(\boldsymbol{\theta})}t_k(\boldsymbol{\theta}) \quad\Longrightarrow\quad \log t_k(\boldsymbol{\theta}) \leftarrow \log t_k(\boldsymbol{\theta}) + (\log q_k(\boldsymbol{\theta}) - \log q_g(\boldsymbol{\theta})). \tag{37}$$

Finally, the server update in Ashman et al. (2022, Alg. 1) is given by,

$$q_g^{(t)}(\boldsymbol{\theta}) \propto q_g^{(t-1)}(\boldsymbol{\theta})\prod_{k=1}^{K}\frac{t_k^{(t)}(\boldsymbol{\theta})}{t_k^{(t-1)}(\boldsymbol{\theta})} \propto \pi_0(\boldsymbol{\theta})\prod_{k=1}^{K}t_k^{(t)}(\boldsymbol{\theta}), \tag{38}$$

where the last step holds when $t_k^{(0)}(\boldsymbol{\theta}) = 1$ and $q_g^{(0)}(\boldsymbol{\theta}) = \pi_0(\boldsymbol{\theta})$, which is the initialization condition in Ashman et al. (2022, Alg. 1). The last step can be seen by recursively applying the definition of $q_g$ and noticing that successive terms in the product cancel out in a telescoping fashion.

We now discuss the difference between Bayesian-ADMM and PVI (Eqs. 36 to 38) in more detail. Unlike PVI, Bayesian-ADMM uses the learning rate $\rho$ in all three updates, whereas PVI does not use any learning rates in its raw form (Ashman et al., 2022). The learning rate is essential for convergence and it is well-known that some splitting algorithms can diverge if learning-rates are not carefully chosen. To get a method that works well in practice, Swaroop et al. (2025) used a learning rate $\rho$ in the dual update with a justification of damping (similarly to Ashman et al. (2022)) but they did not use it in the update of $q_k$.

The server update of PVI in Eq. 38 makes the method more similar to the Alternating Minimization Algorithm (AMA) of Tseng (1991, Eq. 3.4a–3.4c) than ADMM. AMA has the following form:

$$\text{Client updates:} \quad \boldsymbol{\theta}_k \leftarrow \arg\min_{\boldsymbol{\theta}_k} \; \ell_k(\boldsymbol{\theta}_k) + \mathbf{v}_k^\top \boldsymbol{\theta}_k + \frac{\rho}{2} \|\boldsymbol{\theta}_k - \boldsymbol{\theta}_g\|^2, \tag{39}$$

$$\mathbf{v}_k \leftarrow \mathbf{v}_k + \rho(\boldsymbol{\theta}_k - \boldsymbol{\theta}_g), \tag{40}$$

$$\text{Server update:} \quad \boldsymbol{\theta}_g \leftarrow \arg\min_{\boldsymbol{\theta}_g} \; \ell_0(\boldsymbol{\theta}_g) - \sum_{k=1}^{K} \mathbf{v}_k^\top \boldsymbol{\theta}_g. \tag{41}$$

AMA does not use a proximal-term on the server, and for $\ell_0(\boldsymbol{\theta}_g) = \frac{1}{2}\|\boldsymbol{\theta}_g\|^2$ the server update is $\boldsymbol{\theta}_g \leftarrow \sum_{k=1}^{K} \mathbf{v}_k$, which is similar to PVI's server update. However, AMA also uses a step-size in the primal and dual updates, whereas PVI only uses a step-size in the dual, which is referred to as a damping parameter. From the convergence proof of AMA, it is clear that step-sizes are required to obtain an overall convergent method, see Tseng (1991, Eq. 3.4d).

Similar to Bayesian-ADMM, PVI with damping can also be connected to the Bayesian learning rule. Taking the optimality condition of Eq. 36 for an exponential-family $q(\boldsymbol{\theta})$ and inserting it into the dual step Eq. 37 with damping, we get:

$$\boldsymbol{\eta}_k \leftarrow (1-\rho)\boldsymbol{\eta}_k + \rho\nabla_{\boldsymbol{\mu}}\mathbb{E}_{q_k}[-\ell_k]\big|_{\boldsymbol{\mu}=\boldsymbol{\mu}_k}, \tag{42}$$

where we used $t_k(\boldsymbol{\theta}) = \exp(\langle \boldsymbol{\eta}_k, \mathbf{T}(\boldsymbol{\theta})\rangle)$. This is a reformulation of the Bayesian learning rule (Khan & Rue, 2023) in terms of local parameters, see Khan & Rue (2023, Eq. 60); see also Khan & Lin (2017), except that the natural gradients are evaluated at $\boldsymbol{\mu}_k$ instead of $\boldsymbol{\mu}$.

### C.4 Derivation of Bayesian Duality as Stationarity Condition of a Lagrangian

Here, we show how saddle-points of a Lagrangian recover the Bayesian duality structure of Fig. 3 (right) and Eq. 13. Consider the following,

$$\text{Lagrangian}(\boldsymbol{\mu}_k, \boldsymbol{\eta}_k, \boldsymbol{\mu}_g) = \sum_{k=1}^{K} \big[ \mathcal{L}_k(\boldsymbol{\mu}_k) + \langle \boldsymbol{\eta}_k, \boldsymbol{\mu}_k - \boldsymbol{\mu}_g \rangle \big] + \text{KL}(q_g \,\|\, \pi_0). \tag{43}$$

At a saddle-point, all the derivatives with respect to the primal- and dual variables vanish. Taking the derivative with respect to $\boldsymbol{\eta}_k$ and setting it to zero we immediately get $\boldsymbol{\mu}_k^* = \boldsymbol{\mu}_g^*$ from Eq. 13 due to linearity of the inner product. Taking the derivative with respect to $\boldsymbol{\mu}_k$ and setting it to zero directly leads to $\boldsymbol{\eta}_k^* = -\nabla\mathcal{L}_k(\boldsymbol{\mu}_k^*)$. Taking the gradient with respect to $\boldsymbol{\mu}_g$ and setting it to zero, we get

$$\boldsymbol{\lambda}_g^* = \sum_{k=0}^{K} \boldsymbol{\eta}_k^* = -\sum_{k=0}^{K} \nabla\mathcal{L}_k(\boldsymbol{\mu}_g^*),$$

which follows as in App. B.2. The last part in Eq. 13, $\boldsymbol{\mu}_g^* = \nabla A(\boldsymbol{\lambda}_g^*)$, simply follows from the EF's dual map.

The Lagrangian is a direct extension of that used for Eq. 1, where $(\boldsymbol{\theta}_k, \mathbf{v}_k, \boldsymbol{\theta}_g)$ are replaced by $(\boldsymbol{\mu}_k, \boldsymbol{\eta}_k, \boldsymbol{\mu}_g)$. One non-trivial change is the KL term, which makes sense given its presence in the VB objective. The linear terms stem from the constraint $\boldsymbol{\mu}_g = \boldsymbol{\mu}_k$ in the $\boldsymbol{\mu}$-space. This choice of coordinates is natural and used in other works as well (Adam et al., 2021; Opper & Winther, 2005). Exponential families are also derived using such constraints, see App. B.1.

## D Derivation of the ADMM Extensions in Sec. 3.4 and 3.5

### D.1 Recovering Federated ADMM

As mentioned in Sec. 3.4, we recover federated ADMM when using Gaussians with fixed covariance. These form an exponential family as described in Eq. 21, see also Table 3. From Eq. 21, we see that the natural gradients are simply an expectation of the regular gradients. Since $\mathbf{T}(\boldsymbol{\theta}) = \boldsymbol{\theta}$, and the natural gradients are just gradients as in regular ADMM, the dual variable $\boldsymbol{\eta}_k$ has the same

size as in ADMM, that is, we set $\boldsymbol{\eta}_k = \mathbf{v}_k$. As dual variables are the natural gradients, we also get $\mathbf{v}_k = -\mathbb{E}_{q_k}[\nabla \ell_k(\boldsymbol{\theta})]$.

The KL divergence between two Gaussians with fixed covariance is simply the squared Euclidean norm (up to a constant), and with $\boldsymbol{\mu} = \mathbf{m}$, the client update in Bayesian-ADMM in Eq. 15 becomes

$$\mathbf{m}_k \leftarrow \operatorname{argmin}_{\mathbf{m}_k} \mathbb{E}_{\mathcal{N}(\boldsymbol{\theta}|\mathbf{m}_k, \mathbf{I})}[\ell_k(\boldsymbol{\theta})] + \mathbf{v}_k^\top \mathbf{m}_k + \frac{\rho}{2}\|\mathbf{m}_k - \mathbf{m}_g\|^2. \tag{44}$$

Using the definition of the expectation parameter and $\mathbf{T}(\boldsymbol{\theta}) = \boldsymbol{\theta}$, we can write $\mathbf{v}_k^\top \mathbf{m}_k = \mathbb{E}_{q_k}[\mathbf{v}_k^\top \boldsymbol{\theta}]$ and using linearity of the expectation, Eq. 44 then recovers Eq. 22 from the main text. The dual update of ADMM is recovered from Bayesian-ADMM by plugging in $\boldsymbol{\eta}_k = \mathbf{v}_k$, $\boldsymbol{\lambda}_k = \mathbf{m}_k$, $\boldsymbol{\lambda}_g = \mathbf{m}_g$.

The same arguments as for Eq. 44 allow us to write Bayesian-ADMM server step as,

$$\mathbf{m}_g \leftarrow \operatorname{argmin}_{\mathbf{m}_g} \mathbb{E}_{\mathcal{N}(\boldsymbol{\theta}|\mathbf{m}_g, \mathbf{I})}[\ell_0(\boldsymbol{\theta})] - \sum_{k=1}^K \mathbf{v}_k^\top \mathbf{m}_g + \sum_{k=1}^K \frac{\rho}{2}\|\mathbf{m}_g - \mathbf{m}_k\|^2. \tag{45}$$

Using the delta method, we see that this recovers the server update of regular ADMM.

## D.2 Derivation of Newton-like ADMM via Full Covariances

To derive the Newton-like method, we use the following forms for the posterior distributions

$$q_g(\boldsymbol{\theta}) = \mathcal{N}(\boldsymbol{\theta} \,|\, \mathbf{m}_g, \mathbf{S}_g^{-1}), \quad q_k(\boldsymbol{\theta}) = \mathcal{N}(\boldsymbol{\theta} \,|\, \mathbf{m}_k, \mathbf{S}_k^{-1}). \tag{46}$$

These fit our setup (see Table 3) through,

$$\boldsymbol{\lambda} = (\mathbf{Sm}, -\tfrac{1}{2}\mathbf{S}), \qquad \boldsymbol{\mu} = (\mathbf{m}, \mathbf{mm}^\top + \mathbf{S}^{-1}), \qquad \mathbf{T}(\boldsymbol{\theta}) = (\boldsymbol{\theta}, \boldsymbol{\theta}\boldsymbol{\theta}^\top). \tag{47}$$

Following Khan & Rue (2023, Eq.10-11), the natural gradients for Gaussians with full covariance are

$$\nabla_{\boldsymbol{\mu}} \mathbb{E}_{q_k}[\ell_k] = \left(\mathbf{g}_k - \mathbf{H}_k \mathbf{m}_k, \tfrac{1}{2}\mathbf{H}_k\right), \qquad \boldsymbol{\eta}_k = (\mathbf{v}_k, -\tfrac{1}{2}\mathbf{V}_k), \tag{48}$$

where $\mathbf{H}_k = \mathbb{E}_{q_k}[\nabla^2 \ell_k]$ is now the full expected Hessian which determines the form of the second dual variable to be a matrix. Since $\boldsymbol{\eta}_k$ is the negative natural gradient, we get that $\mathbf{V}_k = \mathbf{H}_k$ to be the Hessian. To rewrite the client problem in Eq. 15, we use linearity of expectation and definition of the expectation parameter to get $\langle \boldsymbol{\eta}_k, \boldsymbol{\mu}_k \rangle = \mathbb{E}_{q_k}[\langle \boldsymbol{\eta}_k, \mathbf{T}(\boldsymbol{\theta}) \rangle]$ and expand $\langle \boldsymbol{\eta}_k, \mathbf{T}(\boldsymbol{\theta}) \rangle = \langle (\mathbf{v}_k, -\tfrac{1}{2}\mathbf{V}_k), (\boldsymbol{\theta}, \boldsymbol{\theta}\boldsymbol{\theta}^\top) \rangle = \mathbf{v}_k^\top \boldsymbol{\theta} - \tfrac{1}{2}\boldsymbol{\theta}^\top \mathbf{V}_k \boldsymbol{\theta}$ to arrive at:

$$\mathbf{m}_k, \mathbf{S}_k \leftarrow \operatorname{argmin}_{\mathbf{m}_k, \mathbf{S}_k} \mathbb{E}_{q_k}[\ell_k(\boldsymbol{\theta}) + \mathbf{v}_k^\top \boldsymbol{\theta} - \tfrac{1}{2}\boldsymbol{\theta}^\top \mathbf{V}_k \boldsymbol{\theta}] + \rho \mathrm{KL}(q_k \,\|\, q_g), \tag{49}$$

where the KL-divergence is given as,

$$\mathrm{KL}(q_k \,\|\, q_g) = \tfrac{1}{2}\left(\mathrm{tr}(\mathbf{S}_k^{-1}\mathbf{S}_g) + \log\det(\mathbf{S}_k) + \|\mathbf{m}_k - \mathbf{m}_g\|_{\mathbf{S}_g}^2\right) + \mathrm{const}. \tag{50}$$

Substituting the form of $\boldsymbol{\eta}_k$ and $\boldsymbol{\lambda}_k$ as well as $\boldsymbol{\lambda}_g$, the dual update in Bayesian-ADMM (Eq. 16) is,

$$\mathbf{v}_k \leftarrow \mathbf{v}_k + \rho(\mathbf{S}_k \mathbf{m}_k - \mathbf{S}_g \mathbf{m}_g), \tag{51}$$

$$\mathbf{V}_k \leftarrow \mathbf{V}_k + \rho(\mathbf{S}_k - \mathbf{S}_g). \tag{52}$$

Finally, to write the server update of Bayesian-ADMM (Eq. 17), we first notice that it has a simple closed form in natural parameters. To see this, we take the derivative in $\boldsymbol{\mu}_g$ and set it to zero to get,

$$\boldsymbol{\lambda}_g - \boldsymbol{\eta}_0 - \sum_{k=1}^K \boldsymbol{\eta}_k + \rho \sum_{k=1}^K (\boldsymbol{\lambda}_g - \boldsymbol{\lambda}_k) = 0, \tag{53}$$

where $\boldsymbol{\eta}_0 = -\nabla_{\boldsymbol{\mu}} \mathcal{L}_0(\boldsymbol{\mu}_g)$. Rearranging and denoting $\alpha = 1/(1 + \rho K)$, this can be written as

$$\boldsymbol{\lambda}_g = (1 - \alpha)\frac{1}{K}\sum_{k=1}^K \boldsymbol{\lambda}_k + \alpha \sum_{k=0}^K \boldsymbol{\eta}_k. \tag{54}$$

Using the form of $\boldsymbol{\lambda}$ from Eq. 47, the form of $\boldsymbol{\eta}_k$ from Eq. 48 and a simple isotropic Gaussian prior with precision $\delta$, this can be simplified as:

$$\mathbf{S}_g \leftarrow (1 - \alpha)\,\mathtt{Mean}\,(\mathbf{S}_{1:K}) + \alpha\left[\delta\mathbf{I} + \mathtt{Sum}(\mathbf{V}_{1:K})\right], \tag{55}$$

$$\mathbf{m}_g \leftarrow \mathbf{S}_g^{-1}\left[(1 - \alpha)\mathtt{Mean}(\mathbf{S}_{1:K}\mathbf{m}_{1:K}) + \alpha\mathtt{Sum}(\mathbf{v}_{1:K})\right]. \tag{56}$$

We use this full Newton-like method for the experiments in Figs. 4, 5a and 5b where the variational subproblem in Eq. 49 is solved using VON (Khan & Rue, 2023, Eq. 12).

**Convergence in a single communication round.** Similar to message passing algorithms (Koller & Friedman, 2009; Winn et al., 2005), Bayesian-ADMM can converge in a single round of communication, which is unlike regular ADMM and BregmanADMM (Wang & Banerjee, 2014).

**Proposition D.1.** *If $\ell_k(\boldsymbol{\theta}) = -\langle \mathbf{c}_k, \mathbf{T}(\boldsymbol{\theta}) \rangle$ (for example, $\ell_k(\boldsymbol{\theta}) = \frac{1}{2}\boldsymbol{\theta}^\top \mathbf{A}\boldsymbol{\theta} + \mathbf{b}^\top \boldsymbol{\theta}$ in the case of full Gaussian distributions), step-size $\rho = 1/K$ and initializing the server at the prior $\boldsymbol{\lambda}_g = \boldsymbol{\eta}_0$, then Bayesian-ADMM (Eqs. 15 to 17) converges to the solution of Eq. 9 after one communication round.*

*Proof.* From the optimality condition of Eq. 15, we get that $\boldsymbol{\lambda}_k = \boldsymbol{\lambda}_g + \frac{1}{\rho}\mathbf{c}_k$ as we initialize $\boldsymbol{\eta}_k = 0$. The server update in Bayesian-ADMM takes a simpler form, as derived in Eq. 54,

$$\boldsymbol{\lambda}_g = (1-\alpha)\frac{1}{K}\sum_{k=1}^{K}\boldsymbol{\lambda}_k + \alpha\sum_{k=0}^{K}\boldsymbol{\eta}_k. \tag{57}$$

Noticing that $\boldsymbol{\eta}_k = \mathbf{c}_k$ after the dual update, the server update is:

$$\boldsymbol{\lambda}_g = (1-\alpha)\frac{1}{K}\sum_{k=1}^{K}\left(\boldsymbol{\eta}_0 + \frac{1}{\rho}\mathbf{c}_k\right) + \alpha\left(\boldsymbol{\eta}_0 + \sum_{k=1}^{K}\mathbf{c}_k\right) = \boldsymbol{\eta}_0 + \sum_{k=1}^{K}\mathbf{c}_k, \tag{58}$$

where in the second equality we used $\rho = 1/K$. This is the optimality condition of Eq. 9 for a conjugate prior $\boldsymbol{\eta}_0$. In the second client step, the Lagrange multiplier term cancels out with the loss (since $\boldsymbol{\eta}_k = \mathbf{c}_k$ and $\ell_k(\boldsymbol{\theta}) = -\mathbf{c}_k^\top \mathbf{T}(\boldsymbol{\theta})$), and minimizing the KL leads to $\boldsymbol{\lambda}_k = \boldsymbol{\lambda}_g$, and therefore all clients and the server reach the optimal solution after one communication round. □

## D.3 DERIVATION OF ADAM-LIKE ALG. 1 (IVON-ADMM)

To derive the Adam-like method shown in Alg. 1, we use a Gaussian posterior approximation with diagonal covariance. Denoting $q(\boldsymbol{\theta}) = \mathcal{N}(\boldsymbol{\theta} \,|\, \mathbf{m}, \mathrm{diag}(\mathbf{s})^{-1})$ where $\mathbf{s}$ is the precision vector, similarly to the previous section, we get the following setup:

$$\boldsymbol{\lambda} = (\mathbf{sm}, -\tfrac{1}{2}\mathbf{s}), \qquad \boldsymbol{\mu} = (\mathbf{m}, \mathbf{m}^2 + 1/\mathbf{s}), \qquad \mathbf{T}(\boldsymbol{\theta}) = (\boldsymbol{\theta}, \boldsymbol{\theta}^2). \tag{59}$$

For the natural-gradient we get:

$$\nabla_{\boldsymbol{\mu}}\mathbb{E}_{q_k}[\ell_k] = \left(\mathbf{g}_k - \mathbf{h}_k\mathbf{m}_k, \tfrac{1}{2}\mathbf{h}_k\right). \tag{60}$$

Here, all operations (such as $\mathbf{sm}$) are element-wise. The last equation is from Khan & Rue (2023, Eq. 10-11) where we denote $\mathbf{g}_k = \mathbb{E}_{q_k}[\nabla\ell_k]$ and $\mathbf{h}_k = \mathbb{E}_{q_k}[\mathrm{diag}(\nabla^2\ell_k)]$.

Substituting the definitions of a diagonal Gaussian as an EF (see Table 3 and Eq. 59) into the Bayesian-ADMM updates is a straightforward mechanical calculation similar to the previous section. We denote the dual variables as $\boldsymbol{\eta}_k = (\mathbf{v}_k, -\tfrac{1}{2}\mathbf{u}_k)$ to match the structure of the natural parameter $\boldsymbol{\lambda}_k = (\mathbf{s}_k\mathbf{m}_k, -\tfrac{1}{2}\mathbf{s}_k)$ which is also a Lagrange multiplier (see App. B.1). Unlike $\boldsymbol{\lambda}_k$ which has to be a valid natural parameter, $\boldsymbol{\eta}_k$ can be arbitrary (and also zero), so we do not use $\mathbf{u}_k\mathbf{v}_k$, but rather just $\mathbf{v}_k$ for the first argument.

We expand the linear term in Bayesian-ADMM to get,

$$\langle \boldsymbol{\eta}_k, \mathbf{T}(\boldsymbol{\theta}) \rangle = \langle (\mathbf{v}_k, -\tfrac{1}{2}\mathbf{u}_k), (\boldsymbol{\theta}, \boldsymbol{\theta}^2) \rangle = \mathbf{v}_k^\top \boldsymbol{\theta} - \tfrac{1}{2}\mathbf{u}_k^\top(\boldsymbol{\theta})^2 = \mathbf{v}_k^\top \boldsymbol{\theta} - \tfrac{1}{2}\boldsymbol{\theta}^\top \mathrm{diag}(\mathbf{u}_k)\boldsymbol{\theta}. \tag{61}$$

Inserting this into Eq. 15, we arrive at:

$$\mathbf{m}_k, \mathbf{s}_k \leftarrow \mathrm{argmin}_{\mathbf{m}_k, \mathbf{s}_k} \; \mathbb{E}_{q_k}[\ell_k(\boldsymbol{\theta}) + \mathbf{v}_k^\top \boldsymbol{\theta} - \tfrac{1}{2}\boldsymbol{\theta}^\top \mathrm{diag}(\mathbf{u}_k)\boldsymbol{\theta}] + \rho \mathrm{KL}(q_k \,\|\, q_g), \tag{62}$$

where the KL-divergence is given by:

$$\mathrm{KL}(q_k \,\|\, q_g) = \tfrac{1}{2}\left(\sum_{i=1}^{P}\left[\frac{\mathbf{s}_g^i}{\mathbf{s}_k^i} + \log \mathbf{s}_i\right] + \|\mathbf{m}_k - \mathbf{m}_g\|_{\mathbf{s}_g}^2\right) + \mathrm{const}. \tag{63}$$

Alg. 1 uses IVON (Shen et al., 2024) to minimize Eq. 62. We slightly generalize IVON to handle a general prior and the two Lagrange multipliers. The resulting method is shown in Alg. 2 with modifications over Shen et al. (2024) highlighted in red.

The IVON method in Alg. 2 minimizes the following objective function,

$$\lambda\mathbb{E}_q\left[\ell(\boldsymbol{\theta}) + \mathbf{v}^\top \boldsymbol{\theta} - \tfrac{1}{2}\boldsymbol{\theta}^\top \mathrm{diag}(\mathbf{u})\boldsymbol{\theta}\right] + \mathrm{KL}(q \,\|\, p), \tag{64}$$

where $\ell(\boldsymbol{\theta})$ is a generic loss and $\hat{\nabla}\ell(\boldsymbol{\theta})$ denotes a stochastic gradient. The objective Eq. 64 matches the subproblem in Eq. 62. We provide details on how to use Alg. 2 in the following App. D.4.

---

**Algorithm 2** IVON for minimizing Eq. 64. Hyperparameters $h_0, \alpha_t, \beta_1, \beta_2$ are chosen following Shen et al. (2024, Appendix A). We highlight in red the extra terms due to the Lagrange multipliers.

---

**Inputs:** Loss $\ell$, prior $p(\boldsymbol{\theta}) = \mathcal{N}(\boldsymbol{\theta} \mid \mathbf{m}_p, \boldsymbol{\sigma}_p^2)$, loss scaling $\lambda$, Lagrange multipliers $\mathbf{v}, \mathbf{u}$.
**Initialization:** $\mathbf{m} \leftarrow \mathbf{m}_p, \mathbf{h} \leftarrow h_0, \mathbf{g} \leftarrow 0, \boldsymbol{\delta} \leftarrow 1/(\lambda \boldsymbol{\sigma}_p^2)$
1: **for** $t = 1, 2, \ldots$ **do**
2: $\quad \widehat{\mathbf{g}} \leftarrow \widehat{\nabla}\ell(\boldsymbol{\theta})$, where $\boldsymbol{\theta} \sim \mathcal{N}(\mathbf{m}, \boldsymbol{\sigma}^2)$
3: $\quad \widehat{\mathbf{h}} \leftarrow \widehat{\mathbf{g}} \cdot (\boldsymbol{\theta} - \mathbf{m})/\boldsymbol{\sigma}^2 - \mathbf{u}$
4: $\quad \mathbf{g} \leftarrow \beta_1 \mathbf{g} + (1 - \beta_1)\widehat{\mathbf{g}}$
5: $\quad \mathbf{h} \leftarrow \beta_2 \mathbf{h} + (1 - \beta_2)\widehat{\mathbf{h}} + \frac{1}{2}(1 - \beta_2)^2(\mathbf{h} - \widehat{\mathbf{h}})^2/(\mathbf{h} + \boldsymbol{\delta})$
6: $\quad \mathbf{m} \leftarrow \mathbf{m} - \alpha_t(\mathbf{g} + \mathbf{v} - \mathbf{um} + \boldsymbol{\delta}(\mathbf{m} - \mathbf{m}_p))/(\mathbf{h} + \boldsymbol{\delta})$
7: $\quad \boldsymbol{\sigma} \leftarrow 1/\sqrt{\lambda(\mathbf{h} + \boldsymbol{\delta})}$
8: **end for**
9: **return** $(\mathbf{m}, 1/\boldsymbol{\sigma}^2)$

---

### D.4 DETAILS ON USING IVON (ALG. 2) TO IMPLEMENT IVON-ADMM (ALG. 1)

Here, we provide additional details on how the IVON step in Alg. 1 is implemented in our experiments. As commonly done, we consider a tempered version of the variational-Bayesian problem in Eq. 9,

$$q_g^* = \operatorname*{argmin}_{q_g \in \mathcal{Q}} \sum_{k=1}^{K} \mathbb{E}_{q_g}[\ell_k(\boldsymbol{\theta})] + \tau \mathrm{KL}(q_g \parallel \pi_0), \tag{65}$$

where $\tau > 0$ is a temperature parameter. Dividing the whole objective by $\tau$ does not change the minimizer and gives rescaled client losses $\ell_k(\boldsymbol{\theta})/\tau$ on the original objective Eq. 9.

For the tempered problem, the local client optimisation problem in IVON-ADMM reads as follows,

$$\operatorname*{argmin}_{q_k} \mathbb{E}_{q_k} \left[ \sum_{i=1}^{N_k} \ell_k^{(i)}(\boldsymbol{\theta})/\tau + \mathbf{v}_k^\top \boldsymbol{\theta} - \frac{1}{2}\boldsymbol{\theta}^\top \mathrm{diag}(\mathbf{u}_k)\boldsymbol{\theta} \right] + \rho \mathrm{KL}(q_k \parallel q_g), \tag{66}$$

which when dividing by $\rho$ corresponds to the loss in Alg. 1, where we set $\tau = 1$. $\ell_k^{(i)}$ denote the per-example loss functions and $N_k$ is the number of data examples on the client $k$. We now bring this problem into a form that allows us to directly apply Alg. 2. Dividing by $\rho$ we get:

$$\operatorname*{argmin}_{q_k} \frac{N_k}{\rho \tau} \mathbb{E}_{q_k} \left[ \frac{1}{N_k} \sum_{i=1}^{N_k} \ell_k^{(i)}(\boldsymbol{\theta}) + \frac{\tau}{N_k}\mathbf{v}_k^\top \boldsymbol{\theta} - \frac{\tau}{N_k}\frac{1}{2}\boldsymbol{\theta}^\top \mathrm{diag}(\mathbf{u}_k)\boldsymbol{\theta} \right] + \mathrm{KL}(q_k \parallel q_g). \tag{67}$$

Matching the forms of the above equation with Eq. 64 gives us $\lambda = N_k/(\rho \tau)$, $\mathbf{v} = (\tau/N_k)\mathbf{v}_k$, $\mathbf{u} = (\tau/N_k)\mathbf{u}_k$. For each client subproblem in line 4 in Alg. 1 we are calling Alg. 2 as a subroutine with the following inputs: $\ell_k, (\mathbf{m}_g, \boldsymbol{\sigma}_g^2), \frac{N_k}{\rho \tau}, \frac{\tau}{N_k}\mathbf{v}_k$, and $\frac{\tau}{N_k}\mathbf{u}_k$. Alg. 2 then returns $(\mathbf{m}_k, \mathbf{s}_k)$ which is a minimizer of the client variational objective in Bayesian-ADMM.

## E EXPERIMENTAL DETAILS FOR THE ILLUSTRATIVE EXAMPLES

### E.1 TOY EXAMPLE IN FIG. 4

For the toy example in Fig. 4 we run the standard federated ADMM algorithm (with tuned quadratic regularizer $\frac{\delta}{2}\|\boldsymbol{\theta}\|^2$) and Bayesian-ADMM with full covariances from App. D.2. The dataset consists of the shown points, where we append a single dimension to the features to have a bias present in the linear classifier. We use a binary cross entropy loss (logistic regression) and train with $\delta = 0.2$. The step-sizes are set to $\rho = 0.2$ for both methods.

### E.2 ONE-STEP CONVERGENCE ON RIDGE REGRESSION IN FIG. 5A

The one step convergence plot in Fig. 5a uses the same data as in App. E.3, but we consider a linear regression loss $\ell_k(\boldsymbol{\theta}) = \frac{1}{2}\|\mathbf{X}\boldsymbol{\theta} - \mathbf{y}\|^2$ and regularizer $\ell_0(\boldsymbol{\theta}) = \frac{\delta}{2}\|\boldsymbol{\theta}\|^2$. Note that this fits the assumption

of the one-step convergence result in Prop. D.1 for sufficient statistics $\mathbf{T}(\boldsymbol{\theta}) = (\boldsymbol{\theta}, \boldsymbol{\theta}\boldsymbol{\theta}^{\top})$. We again compare the standard ADMM method to our Bayesian-ADMM with full Gaussian covariances as well as a direct application of BregmanADMM (Wang & Banerjee, 2014) which uses $\boldsymbol{\mu}$-coordinates in the dual update of $\boldsymbol{\eta}_k$.

### E.3 Logistic Regression Convergence Plot in Fig. 5b

The setup is a multiclass Bayesian logistic regression problem on MNIST. We have 10 classes, split across $K = 5$ clients where the data is split up in a heterogeneous way: [(0,1), (2,3), (4,5), (6,7), (8,9)]. We run PVI, PVI with damping (Ashman et al., 2022; Swaroop et al., 2025), BregmanADMM (Wang & Banerjee, 2014) and Bayesian-ADMM for full Gaussian posteriors. BregmanADMM is implemented similarly to Bayesian-ADMM but using the $\boldsymbol{\mu}$-coordinates in the dual update. BregmanADMM is applicable to our setting, since the KL-divergence between two exponential families is a Bregman divergence. The client subproblem is solved by running the Variational Online Newton method (Khan & Rue, 2023, Eq. 12) until convergence. PVI with damping uses $\rho = 1/K$ on the dual update (larger $\rho$ did not converge, smaller were slower), whereas for our method we set $\rho = 1$.

## F Hyperparameters, Architectures and Datasets

We provide further details about the hyperparameters, model architectures and datasets used in the experiments in Tables 1 and 2.

For the hyperparameters in IVON-ADMM we perform a coarse grid search over the step-sizes $\rho$ and $\gamma$ in Alg. 1, the prior precision $\delta$ and the temperature $\tau$. Compared to FedProx, there are only two additional hyperparameters (the dual step-size and the temperature). We found the choice $0.1$ to work well for both and we did not further tune them. The subproblem is solved using the IVON optimizer outlined in Alg. 2 whose hyperparameters we set according to the recommendations given in Shen et al. (2024, Appendix A).

For all baseline methods, we follow the hyperparameter tuning procedure from Swaroop et al. (2025). We also ensure that the random dataset splits for our experiments match those from the reported results in Swaroop et al. (2025).

**Details on heterogeneous splits.** We follow the heterogeneous sampling procedure from Swaroop et al. (2025, App E1) to generate heterogeneous splits of MNIST, FashionMNIST and CIFAR-10. When there are 10 clients, 90% of all data is usually within 6 clients, with 2 clients having 50%. Within each client, usually 60-95% of client data belongs to just 4 classes.

### F.1 MNIST

We train on the full MNIST dataset of 60000 examples. All methods use batch-size 32. The model is a fully connected neural network with sigmoid activation functions and two hidden layers with 200 and 100 neurons, respectively.

Results for baselines are taken from Swaroop et al. (2025) as our setup is identical to theirs.

For the highly-heterogeneous 100-client setting, we use the same split as in McMahan et al. (2016), where each client has data from 2 classes only (300 examples from each class in each client). This is a particularly difficult setting as each client only has data from 2 classes. We perform hyperparameter sweeps over baselines following Swaroop et al. (2025, App E4), except we also allow each hyperparameter to be one order of magnitude smaller (as there are more clients).

### F.2 FashionMNIST

For the 10 client splits, we train on $10\%$ of the FMNIST dataset. The 100 client split uses the full dataset. All methods use batch-size 32, and the model is again the same fully connected neural network as for MNIST.

Results for baselines are taken from Swaroop et al. (2025) as our setup is identical to theirs.

Table 4: Additional results showing test accuracy and test NLL for 10, 25 and 50 rounds, with mean and standard deviations over 3 runs. IVON-ADMM significantly outperforms all baselines. Averaging over the posterior in IVON-ADMM (as opposed to IVON-ADMM@$\mathbf{m}_g$) often improves performance.

| Scenario | Method | Test accuracy (↑ larger is better) | | | Test NLL (↓ smaller is better) | | |
|---|---|---|---|---|---|---|---|
| | | 10 rounds | 25 rounds | 50 rounds | 10 rounds | 25 rounds | 50 rounds |
| MLP, 10 clients heterog. 10% FMNIST | FedAvg/FedProx | 69.9±0.4 | 74.7±0.6 | 76.9±0.9 | 0.80±0.04 | 0.71±0.03 | 0.66±0.03 |
| | FedDyn | 73.0±0.6 | 74.6±0.4 | 74.6±0.5 | 0.75±0.04 | 0.70±0.03 | 0.77±0.05 |
| | FedLap | 71.3±0.9 | 74.3±0.4 | 77.6±0.7 | 0.75±0.04 | 0.71±0.06 | 0.65±0.05 |
| | FedLap-Cov | 74.6±0.7 | 78.3±1.0 | 80.5±0.6 | 0.70±0.04 | 0.63±0.04 | 0.60±0.04 |
| | IVON-ADMM@$\mathbf{m}_g$ | **77.0**±0.8 | **81.4**±0.4 | **82.1**±0.1 | **0.65**±0.02 | **0.53**±0.01 | **0.51**±0.00 |
| | IVON-ADMM | **77.0**±0.8 | **81.5**±0.5 | **82.3**±0.2 | **0.65**±0.02 | **0.52**±0.01 | **0.50**±0.00 |

### F.3 CIFAR-10

We train on the full CIFAR-10 dataset. We consider two models, one is a convolutional network used in Swaroop et al. (2025) (from Zenke et al. (2017)) which uses dropout and the other model is the convolutional network used in Acar et al. (2021) which does not use dropout and has fewer convolutional but more fully connected layers. All methods use batch-size 64.

Results for baselines for the first CNN are taken from Swaroop et al. (2025) as our setup is identical to theirs. For the second CNN, we rerun hyperparameter sweeps with similar settings to Swaroop et al. (2025, App E6). In addition to those parameters, we also sweep over Adam learning rate ($10^{-3}$ or $10^{-4}$). We also try SGD with learning rate 0.1 and sweep over learning rate decay (0.992 or 1), so that we follow the hyperparameter sweep settings from Acar et al. (2021).

### F.4 CIFAR-100

We train on the full CIFAR-100 dataset, split across 10 clients using the same Dirichlet parameters. We use a ResNet-20 model (He et al., 2016) which has around 250k parameters. All methods use batch-size 64. No data augmentation is used.

We use the same hyperparameter sweeps as in Swaroop et al. (2025) for CIFAR-10. We note that FedLap-Cov is now prohibitively slow, since computing the diagonal Laplace approximation estimates of the covariance every communication round takes a long time, and so we do not provide results for it.

## G ADDITIONAL RESULTS

The additional results shown in Table 4 confirm the findings from the main paper. IVON-ADMM gets the highest accuracy and lowest test-loss across communication rounds, improving significantly also over the recent previous state-of-the-art Bayesian federated learning FedLap-Cov which uses more expensive Laplace approximations.

We also perform various ablation studies in Bayesian-ADMM and IVON-ADMM.

### G.1 ABLATION: PVI WITH IVON

Here, we compare IVON-ADMM to PVI with damping (Ashman et al., 2022) implemented with IVON (Shen et al., 2024). The resulting method, which we call IVON-PVI is a special case of IVON-ADMM, where the client step size is set to $\rho = 1$ and one uses $\alpha = 1$ in the server update. We use a finely tuned dual damping, as there are no other step-size hyperparameters. We run on ResNet-20 on CIFAR-100, as in the main paper. The results are summarized in the following Table 5.

We see that using the Bayesian-ADMM update gives an advantage over PVI in the early rounds, while reaching comparable accuracies at the end. A benefit of PVI is that there are no step-sizes to tune other than the dual damping. Finally, we note that PVI has not been implemented so far with state-of-the-art methods like IVON, and IVON-PVI is a special case of our IVON-ADMM.

Table 5: Test accuracy and NLL after 25, 50 and 100 rounds, with mean and standard deviations over 3 runs. IVON-ADMM has faster convergence in early rounds than IVON-PVI, but they reach overall comparable accuracies after 100 rounds.

| Scenario | Method | Test accuracy (↑ larger is better) | | | Test NLL (↓ smaller is better) | | |
|---|---|---|---|---|---|---|---|
| | | 25 rounds | 50 rounds | 100 rounds | 25 rounds | 50 rounds | 100 rounds |
| ResNet-20, | IVON-PVI@$\mathbf{m}_g$ | 31.5±0.5 | 41.2±0.6 | **47.4±0.5** | 2.7±0.0 | 2.2±0.0 | **2.0±0.0** |
| 10 clients, | IVON-PVI | 31.5±0.5 | 41.2±0.6 | **47.5±0.5** | 2.7±0.0 | 2.2±0.0 | **2.0±0.0** |
| heterog. CIFAR-100 | IVON-ADMM@$\mathbf{m}_g$ | **40.0±1.0** | **46.2±0.4** | 46.5±0.6 | **2.3±0.0** | **2.1±0.0** | 2.3±0.1 |
| | IVON-ADMM | **40.0±1.0** | **46.2±0.4** | 46.6±0.6 | **2.3±0.0** | **2.1±0.0** | 2.2±0.1 |

## G.2 ABLATION: SENSITIVITY TO INVERSE CLIENT STEP SIZE $\rho$ AND TEMPERATURE $\tau$

We further study the sensitivity of IVON-ADMM to $\rho$ and $\tau$. We again use the ResNet-20 on CIFAR-100 and highlight the parameters used in the main paper.

The results for varying temperature $\tau$ are summarized in Table 6.

Table 6: Temperature ablation for IVON-ADMM. Test accuracy and NLL after 25, 50 and 100 rounds, with mean and standard deviations over 3 runs.

| Scenario | $\tau$ | Test accuracy (↑ larger is better) | | | Test NLL (↓ smaller is better) | | |
|---|---|---|---|---|---|---|---|
| | | 25 rounds | 50 rounds | 100 rounds | 25 rounds | 50 rounds | 100 rounds |
| | 0.02 | 38.6±0.8 | 23.9±9.4 | 25.5±7.6 | 2.5±0.1 | 5.3±1.4 | 4.1±0.1 |
| | 0.05 | **42.0±0.6** | 42.0±0.6 | 41.1±0.6 | **2.2±0.0** | 2.7±0.1 | 2.9±0.1 |
| | 0.1 | 39.9±1.2 | **46.1±0.6** | 46.6±0.8 | 2.3±0.0 | **2.1±0.0** | 2.2±0.1 |
| ResNet-20, | 0.2 | 35.7±1.0 | 45.1±1.0 | **47.9±0.7** | 2.5±0.1 | **2.1±0.0** | **2.0±0.0** |
| 10 clients, | 0.5 | 27.1±1.1 | 36.6±1.6 | 40.2±1.6 | 2.9±0.1 | 2.5±0.1 | 2.3±0.1 |
| heterog. CIFAR-100 | 1 | 20.0±1.2 | 27.5±1.4 | 30.4±1.2 | 3.3±0.1 | 2.9±0.1 | 2.8±0.1 |

Smaller temperatures lead to faster convergence, but the overall best final accuracy is reached for $\tau = 0.2$. Larger temperatures tend to degrade the performance. All experiments in the paper use $\tau = 0.1$ (highlighted as a gray row in the table), but as we can see, an additional tuning of $\tau$ can further improve the results.

The results for varying client inverse step-size $\rho$ are summarized in the following Table 7.

Table 7: Client inverse step-size $\rho$ ablation for IVON-ADMM. Test accuracy and NLL after 25, 50 and 100 rounds, with mean and standard deviations over 3 runs.

| Scenario | $\rho$ | Test accuracy (↑ larger is better) | | | Test NLL (↓ smaller is better) | | |
|---|---|---|---|---|---|---|---|
| | | 25 rounds | 50 rounds | 100 rounds | 25 rounds | 50 rounds | 100 rounds |
| | 0.1 | 1.0±0.0 | 1.0±0.0 | 1.0±0.0 | diverged | diverged | diverged |
| | 0.2 | 26.7±10.6 | 14.3±10.8 | 14.0±10.6 | diverged | diverged | diverged |
| | 0.5 | **39.8±1.2** | **46.2±0.8** | 46.6±0.9 | **2.3±0.0** | **2.1±0.0** | 2.2±0.1 |
| ResNet-20, | 1.0 | 31.4±0.6 | 40.6±0.7 | 45.2±0.6 | 2.7±0.0 | 2.3±0.0 | **2.1±0.0** |
| 10 clients, | 2.0 | 25.4±0.7 | 32.6±0.7 | 38.9±0.7 | 3.0±0.0 | 2.7±0.0 | 2.4±0.0 |
| heterog. CIFAR-100 | 5.0 | 18.2±0.9 | 23.7±0.7 | 29.0±0.8 | 3.4±0.0 | 3.1±0.0 | 2.9±0.0 |

Smaller $\rho$, which gives the client more freedom to deviate from the server's consensus solution, leads to instabilities. Larger $\rho$ constrains the client to be closer to the server's solution, and while leading to a stable convergence degrades the overall performance. The value of $\rho = 0.5$ which we used in the main paper is at a sweet-spot, balancing this trade-off.

