# OpenReview forum: "Federated ADMM from Bayesian Duality"
_ICLR.cc/2026/Conference — ICLR 2026 Poster_

### Official Review · Reviewer_WVuS · 2025-11-02

**Soundness:** 3
**Presentation:** 2
**Contribution:** 3
**Rating:** 6
**Confidence:** 3

**Summary:**

The paper proposes a novel method that combines Bayesian variational inference with the Alternating Direction Method of Multipliers (ADMM) for federated learning scenarios. Based on the concept of "Bayesian Duality," the authors extend traditional parameter optimization to distribution optimization, developing the BayesADMM algorithm. This method not only encompasses standard federated ADMM as a special case (when using fixed-variance Gaussian distributions) but also naturally extends to more complex distributions, producing update rules similar to Newton's method. The authors further derive the IVON-ADMM variant, a computationally efficient implementation suitable for deep learning models. Experimental results show that this method outperforms existing baselines on multiple datasets (MNIST, FashionMNIST, and CIFAR-10), particularly excelling in heterogeneous data distribution scenarios.

**Strengths:**

1. The paper proposes a novel theoretical framework that integrates Bayesian variational inference with the ADMM algorithm, providing a unified perspective for federated learning. This approach of connecting optimization algorithms with probabilistic inference has theoretical depth.
2. The authors clearly demonstrate how traditional federated ADMM serves as a special case of BayesADMM under specific conditions. This theoretical connection provides a new perspective for understanding existing methods and naturally leads to more powerful extensions.
3. The implementation details of IVON-ADMM showcase a practical and efficient algorithm variant. Compared to existing federated learning optimizers (such as FedAvg and FedAdam), it incurs limited additional computational overhead while delivering significant performance improvements.
4. The experimental design is comprehensive, covering both homogeneous and heterogeneous data distribution scenarios, and validates the method's effectiveness across multiple standard datasets. Particularly, Figure 3(b) demonstrates BayesADMM's property of converging in just one communication round for certain loss functions, which is a compelling empirical result.
5. The authors situate their method within the broader context of optimization and Bayesian inference literature, clearly highlighting connections and distinctions with prior works such as Partitioned Variational Inference (PVI) and Bregman ADMM.

**Weaknesses:**

1.Although the paper claims BayesADMM is theoretically superior, it lacks detailed analysis of computational complexity. Specifically, how do the algorithm's computational and communication costs scale with model parameter size when using more complex distributions (such as full-covariance Gaussians)? This is crucial in practical federated learning scenarios.
2.The experimental section only reports average performance without providing standard deviations or statistical significance tests, making it difficult to assess result reliability. Given the stochastic nature of federated learning (such as client selection and data partitioning), such statistical analysis is particularly necessary.
3.Table 4 shows IVON-ADMM's performance across multiple rounds but does not analyze the trade-off between convergence speed and communication rounds. In practical applications, early stopping may be more useful, but the paper does not explore performance comparisons of different methods under limited communication budgets.
4.While the authors mention the concept of "Bayesian Duality," their explanation of its theoretical foundation is not sufficiently deep. The derivation of the BayesADMM algorithm in Section 3.3 is relatively brief, and certain key steps (such as the transition from Equation 26 to 27) lack detailed explanation, which may affect reader comprehension.
5.The paper does not adequately discuss the impact of hyperparameter selection, particularly the sensitivity of regularization parameter ρ and temperature parameter τ. Although the appendix mentions hyperparameter search, it lacks systematic analysis of how these parameters affect algorithm performance.
6.The comparison with recent state-of-the-art federated learning methods (such as Scaffold and FedProx) is not comprehensive. While the paper compares with FederatedADMM and BregmanADMM, these methods are relatively outdated, and more comparisons with recent works should be included.

**Questions:**

1.In practical federated learning scenarios, clients often have different computational capabilities and communication bandwidths. How does BayesADMM adapt to this system heterogeneity? Specifically, what is the algorithm's robustness when certain clients cannot complete the full variational inference update?
2.From Figure 3(a), it can be seen that PVI diverges without damping while BayesADMM remains stable. Could you provide a detailed analysis of BayesADMM's convergence guarantees, particularly theoretical guarantees for non-convex optimization problems?
3.IVON-ADMM uses diagonal covariance approximation in its implementation, which may lead to information loss in deep learning models. How much performance improvement do you think using low-rank approximation or other covariance structures would bring? How much would the computational overhead increase?
4.In Algorithm 1, you mention "implementation details in Appendix G," but the provided PDF excerpt does not contain this section. Could you briefly explain the key implementation differences between IVON-ADMM and standard ADMM, particularly the techniques used when handling high-dimensional parameter spaces?
5.The paper states that BayesADMM can converge in just one communication round for certain loss functions. Is this property limited to specific types of loss functions? Could you provide more general conditions that specify when the algorithm can converge quickly?

---

> ### Author Response · Authors · 2025-11-21
> **Thanks for your review!**
>
> We thank the reviewer for their constructive feedback, and appreciate that the reviewer thought the work is novel theoretically, and performs well empirically.
>
> > Q1. Although the paper claims BayesADMM is theoretically superior, it lacks detailed analysis of computational complexity. Specifically, how do the algorithm's computational and communication costs scale with model parameter size when using more complex distributions (such as full-covariance Gaussians)? This is crucial in practical federated learning scenarios.
>
> A1. We have an analysis of the computational and communication complexity in the introduction and main paper (lines 357-362), diagonal Gaussians double the communication costs (from P to 2P, where P is the number of parameters). Full Gaussians will move communications costs from P (regular ADMM) to P + P*(P-1)/2. In practice, most of the computation time is spent in backward passes during local training, so computation complexity is the same as other methods (eg FedDyn), which we have now addressed in lines 359-360. The explicit runtimes for the CIFAR-100 ResNet-20 experiment are the same for every method (around ~12h) when using 25 epochs on each client over 100 communication rounds. For the final version, we will include precise measurements.
>
> > Q2.The experimental section only reports average performance without providing standard deviations or statistical significance tests, making it difficult to assess result reliability. Given the stochastic nature of federated learning (such as client selection and data partitioning), such statistical analysis is particularly necessary.
>
> A2. This is incorrect. All the real-world benchmark results have error bars (3 random seeds).
>
> > Q3. Table 4 shows IVON-ADMM's performance across multiple rounds but does not analyze the trade-off between convergence speed and communication rounds. In practical applications, early stopping may be more useful, but the paper does not explore performance comparisons of different methods under limited communication budgets.
>
> A3. We do not understand: we do have results under different communication rounds. Please clarify further what could be added to Table 1. We explore comparisons under limited communication budgets. This is why we provide performance after multiple rounds for all our main results. We see that IVON-ADMM performs best if the number of communication rounds is limited to 10, or 25, or 50 rounds.
>
> > Q4. While the authors mention the concept of "Bayesian Duality," their explanation of its theoretical foundation is not sufficiently deep. The derivation of the BayesADMM algorithm in Section 3.3 is relatively brief, and certain key steps (such as the transition from Equation 26 to 27) lack detailed explanation, which may affect reader comprehension.
>
> A4. We provide a detailed explanation for our derivations. For example, the transition from Equation 27 to 28 (26 to 27 in old version)  is provided immediately after in line 894 (it uses the linearity of the inner product and the definition of expectation parameter). We can add additional explanations for this or for other steps, please let us know.
>
> > Q5. The paper does not adequately discuss the impact of hyperparameter selection, particularly the sensitivity of regularization parameter ρ and temperature parameter τ. Although the appendix mentions hyperparameter search, it lacks systematic analysis of how these parameters affect algorithm performance.
>
> A5. We have added an ablation study for sensitivity of rho and temperature tau to App. J2 and also mention it in the main text (lines 482-485).
>
> > Q6. The comparison with recent state-of-the-art federated learning methods (such as Scaffold and FedProx) is not comprehensive. While the paper compares with FederatedADMM and BregmanADMM, these methods are relatively outdated, and more comparisons with recent works should be included.
>
> A6. We compare with state-of-the-art federated learning methods: in Table 1 and 2 we compare BayesADMM to FedProx and FedDyn (which is much better than Scaffold as shown in the FedDyn paper) and FedLap (ICLR 2025).  Which other methods would you want to see additionally?

---

> ### Author Response · Authors · 2025-11-21
>
> > Q8. From Figure 3(a), it can be seen that PVI diverges without damping while BayesADMM remains stable. Could you provide a detailed analysis of BayesADMM's convergence guarantees, particularly theoretical guarantees for non-convex optimization problems?
>
> A8. Convergence analysis of BayesADMM is an open problem in optimization and would constitute a paper on its own (see also detailed response to Reviewer B61X, Q5).
>
> > Q9. IVON-ADMM uses diagonal covariance approximation in its implementation, which may lead to information loss in deep learning models. How much performance improvement do you think using low-rank approximation or other covariance structures would bring? How much would the computational overhead increase?
>
> A9. The trade-off between more complex posterior approximations and communication overheads is interesting to study. There do not exist reliable methods for non-diagonal posterior estimates in deep learning, which is why we mostly study the diagonal setting. It is therefore unclear how large the performance gain will be, but as we see, the diagonal setting already provides significant improvements.
>
> > Q10. In Algorithm 1, you mention "implementation details in Appendix G," but the provided PDF excerpt does not contain this section. Could you briefly explain the key implementation differences between IVON-ADMM and standard ADMM, particularly the techniques used when handling high-dimensional parameter spaces?
>
> A10. We are confused, as Appendix G is in the PDF that we can access from openreview. Appendix G explains how the subproblem can be solved using the IVON optimizer in Algorithm 2. The rest of the differences are discussed in Algorithm 1 (highlighted in red) and Section 3.5 (where we discuss Algorithm 1). We furthermore will open-source our implementation upon acceptance and have attached the code as a supplementary material.
>
> > Q11. The paper states that BayesADMM can converge in just one communication round for certain loss functions. Is this property limited to specific types of loss functions? Could you provide more general conditions that specify when the algorithm can converge quickly?
>
> A11. Proposition 3.2 analyzes the class of loss functions for which BayesADMM converges in a single communication round. It is indeed limited to a specific type of loss, that depends on the posterior approximation. General conditions and convergence rates are open problems for future research.

---

### Official Review · Reviewer_DtMH · 2025-11-09

**Soundness:** 2
**Presentation:** 3
**Contribution:** 2
**Rating:** 2
**Confidence:** 4

**Summary:**

The paper proposes a general Bayesian perspective on federated ADMM. Under this framework, the classical ADMM emerges as a special case corresponding to isotropic Gaussian posteriors, while more expressive exponential-family posteriors yield new variants. One such variant called _IVON-ADMM_ is derived using diagonal Gaussian covariance and is claimed to perform better in heterogeneity and uncertainty cases of federated learning.

**Strengths:**

__Conceptual novelty:__ The paper establishes a novel Bayesian duality perspective that unifies ADMM and VB under a single framework. This is an interesting connection that could inspire extensions of primal-dual optimization methods.

__Clear motivation and exposition:__ The introduction and backgrounds are well written and clearly position the work relative to prior ADMM and PVI approach (Swaroop et al., 2025).

__Framework generality:__ The proposed Bayesian duality formulation provides a principled way to derive new ADMM-like algorithms by changing the exponential-family posterior.

__Readable presentation:__ For the most part, the paper is well structured and logically progress from classical ADMM to its Bayesian interpretation and finally to _IVON-ADMM_.

**Weaknesses:**

__Soundness of the Formulation:__ While the high-level idea is promising, the derivation in section 3.3 raises concerns about mathematical consistency:
- The "Bayesian ADMM" updates (Eqns. 12-14) are expected to follow from alternating optimization of the Lagrangian in Eqn. 11. However, the replacement of the dual update term $\mu_k - \bar{\mu}$ with $\lambda_k - \bar{\lambda}$ lacks justification within the Lagrangian formulation. The reasoning provided in Appendix E.2, appealing to Bayesian intuition, seems heuristic rather derivational.
- Equation 11 itself may need reconsideration: shifting the linear term $<\hat{\lambda}_k, \mu_k - \hat{\mu}>$ between the sub problems while keeping others fixed breaks symmetry between local and global updates, potentially undermining the claimed equivalence to ADMM.
- Overall, the theoretical grounding of the "Bayesian duality" remains somewhat fragile: the proposed updates look reasonable by analogy but are not rigorously shown to correspond to valid saddle-point dynamics of the stated objective.

__Relation to Existing Work:__
- The method appears to be a straightforward extension of _PVI_ with modified update equations. The novelty over _PVI_ is mainly the introduction of the step size $\rho$ and reinterpretation of dual variables. The paper should make a stronger argument for why this constitutes a _fundametal_ new framework rather  than a variant of _PVI_ with heuristic scaling.
A direct empirical or theoretical  comparison with _PVI_ (as Eqn. 4) is missing. Including  such results would make the claimed advantages more credible.


__Experimental Evaluation__:  The experiments, while broad, are not yet conclusive about the claimed benefits in heterogeneity and uncertainty.
- Figure 3-4 provide illustrative  but small-scale toy examples; they show qualitative improvement but not a clear quantitative advantage.
- The key claim that _IVON_ handles heterogeneity better by leveraging posterior covariance is not substantiated with ablation or analysis showing the role of uncertainty.
- Comparisons with both BayesADMM (without _IVON_) and _PVI_ are missing. Including them would help isolate what _IVON_ adds.
- The computational overhead relative to FedDyn should be quantified to see the computation gain compared to the performance gain.

__Clarity and Notation:__ Several presentation issues reduce readability and reproducibility.
- Step 3 of Fig. 2 is valid only for $\alpha=\frac{1}{1+\rho K}$.
- Many symbols and methods are used before being introduced:
    + BLR first appears on lines 294 and 297 without citation.
    + $q_{1:K}, \hat{t}_{1:K},\bar{q}$ are used before definition in line 215.
- The discussion of natural vs. ordinary gradients is confusing. The paper should use distinct and properly defined notations for both.
- Section 3.2  could be organized better: the correspondence between Eqns. 2 and 10 is conceptually interesting but presented unclearly, with inconsistent references to $\hat{t}_k, \lambda_k, \text{ and } \mu_k$.

**Questions:**

1. What specific mechanism makes BayesADMM or IVON-ADMM handle heterogeneity and uncertainty better than existing method? Can you clarify the role of the posterior covariance in this improvement and demonstrate it experimentally?
2. In Fig. 4, how are the gray uncertainty contours generated? Are they derived from posterior covariance?
3. Why does the exposition (Eqns. 1, 2, 10, 11) rely on the plain Lagrangian, while the ADMM's implementation (Eqns. 3, 12-14) uses the augmented Lagrangian version?
4. The notation $\hat{\lambda}_k$ seems to play dual roles -- as natural parameters of site functions in Sec. 3.2 and as dual variables in Sec. 3.3. Are both interpretations valid? If so, explain their precise connection.

---

> ### Author Response · Authors · 2025-11-21
> **Thank you for your review!**
>
> We thank the reviewer for their time and for acknowledging strengths such as novelty, motivation and readability. We respond to weaknesses below. We have also posted an updated version (changes in blue) which we believe addresses all comments and concerns. We would appreciate it if the reviewer will consider increasing their score.
>
> > Q1.1: The "Bayesian ADMM" updates (Eqns. 12-14) are expected to follow from alternating optimization of the Lagrangian in Eqn. 11. However, the replacement of the dual update term lacks justification within the Lagrangian formulation. The reasoning provided in Appendix E.2, appealing to Bayesian intuition, seems heuristic rather than derivational.
>
> > Q1.2: Overall, the theoretical grounding of the "Bayesian duality" remains somewhat fragile: the proposed updates look reasonable by analogy but are not rigorously shown to correspond to valid saddle-point dynamics of the stated objective.
>
> > Q1.3: Why does the exposition (Eqns. 1, 2, 10, 11) rely on the plain Lagrangian, while the ADMM's implementation (Eqns. 3, 12-14) uses the augmented Lagrangian version?
>
> A1: There is a misunderstanding. We do not do alternating optimization of the Lagrangian (from Eq 11), and our method is not derived from the Lagrangian. It seems that our writing was confusing and so we have now removed the Lagrangian in the updated version. Put in another way, we extend ADMM to variational versions, that is, Eq 3b to Eq 14.
>
> Our main contribution is Eq 10, which is the duality structure underlying the variational objective: we highlight the duality in the structure of \lambda and \mu, which has not been shown before, eg in PVI or Swaroop et al. (2025): we therefore fix the issues in these previous works.
>
> Our method follows valid saddle-point dynamics and it is easy to see that fixed points are stationary points of the saddle-point objective in Appendix D.
>
> > Q2: Equation 11 itself may need reconsideration: shifting the linear term between the sub problems while keeping others fixed breaks symmetry between local and global updates, potentially undermining the claimed equivalence to ADMM.
>
> A2: We do not understand, can you please clarify? The equivalence to ADMM when using isotropic Gaussians is proven, and the exact proposed updates are the correct ones. ADMM does the same thing with the <v_k, theta_k - \bar theta> term in Equation 1. Regardless, as we said in our previous answer, we have removed Equation 11 from the main paper in order to avoid confusions regarding the Lagrangian.
>
> > Q3: The method appears to be a straightforward extension of PVI with modified update equations. The novelty over PVI is mainly the introduction of the step size  and reinterpretation of dual variables. The paper should make a stronger argument for why this constitutes a fundamental new framework rather than a variant of PVI with heuristic scaling. A direct empirical or theoretical comparison with PVI (as Eqn. 4) is missing. Including such results would make the claimed advantages more credible.
>
> A3: We improve upon PVI and have additional experiments to highlight this.
>
> (i) We show an empirical comparison to PVI in Figure 3(a), showing the improvements on a smaller example; (ii) FedLap and FedLap-Cov are based on PVI (but with Laplace approximations), and we show improvements compared to both methods in Tables 1 and 2; (iii) We have also now added an empirical comparison to PVI+IVON (which does not use a Laplace approximation) in Figure 5 (ResNet-20 on CIFAR-100) and Appendix J.1, showing how IVON-ADMM is better.
>
> We find the sentiment that our method is ‘a straightforward extension’ of PVI a little harsh. We show fundamental new connections to AMA (Tseng et al., 1991) and ADMM, and this leads to a specific new algorithm. In our view, it may only seem straightforward in hindsight, which many good papers do (for example, the Adam optimizer is a ‘straightforward extension’ of RMSprop).
>
> > Q4: Figure 3-4 provides illustrative but small-scale toy examples; they show qualitative improvement but not a clear quantitative advantage.
>
> A4: Yes, Figures 3 and 4 give an intuition why BayesADMM works better, while Tables 1 and 2 show the clear quantitative advantage in many larger settings.

---

> ### Author Response · Authors · 2025-11-21
>
> > Q6: The computational overhead relative to FedDyn should be quantified to see the computation gain compared to the performance gain.
>
> A6: The computation is dominated by the cost of backward passes, and therefore the computation cost of IVON-ADMM is the same as FedDyn; we have made this clearer in line 360. The explicit runtimes for the CIFAR-100 ResNet-20 experiments when using 25 epochs on for each client and 100 communication rounds are all around 12 hours for FedAvg/FedProx/FedDyn/FedLap and IVON-ADMM. For the final version, we will add precise measurements of total runtimes.
>
> > Q7: Step 3 of Fig. 2 is valid only for alpha = …
>
> A7: We disagree – primal-dual algorithms can be used with different learning rates for primal and dual updates, and it is common in ADMM literature to use different learning rates for each step. See for instance,
> Chambolle & Pock, A first order primal-dual algorithm for convex problems with applications to imaging.
> Proximal ADMM methods, e.g., as presented in Yin & Ryu, Large-Scale Convex Optimization: Algorithms & Analyses via Monotone Operators.
>
> > Q8: Many symbols and methods are used before being introduced: BLR first appears on lines 294 and 297 without citation. [...] are used before definition in line 215. The discussion of natural vs. ordinary gradients is confusing. The paper should use distinct and properly defined notations for both.
>
> A8: Thanks! We have fixed some of these in the final version. Natural gradients \tilde \nabla follows the standard notation used in literature, e.g., as used by Amari (https://ieeexplore.ieee.org/document/6790500).
>
> > Q9: Section 3.2 could be organized better: the correspondence between Eqns. 2 and 10 is conceptually interesting but presented unclearly, with inconsistent references to t_k, lambda_k and mu_k
>
> A9: We are unsure what the reviewer means by inconsistent references. Could you perhaps elaborate? We are happy to improve the presentation in a final version.
>
> > Q10: In Fig. 4, how are the gray uncertainty contours generated? Are they derived from posterior covariance?
>
> A10: Yes, they are obtained from the posterior covariance in the following way: We compute the Bayesian model predictive distribution p(f|y) by marginalizing out the parameter-space posterior and showing where p(f | y) is larger than 25% (see line 99 in plot_illustration.py in the attached code).
>
> > Q11: The notation \hat lambda_k seems to play dual roles -- as natural parameters of site functions in Sec. 3.2 and as dual variables in Sec. 3.3. Are both interpretations valid? If so, explain their precise connection.
>
> A11: Both interpretations are correct, and it is this unknown connection that we exploit in our method. Note that site functions do not have natural parameters per se, because they may not be normalizable: instead, they have dual parameters \hat\lambda. These are equal to the sum of natural gradients of data in our paper, which allows the site functions to have normalizable dual parameters, which means that we can call them natural parameters. The precise connection is in Eq 9. The approximate posterior q is a product of site functions.
>
> We believe that this connection (that the same parameters are dual variables and natural parameters of the site functions) is what allows us to connect Bayesian methods to ADMM!

---

### Official Review · Reviewer_NFZi · 2025-11-10

**Soundness:** 3
**Presentation:** 3
**Contribution:** 2
**Rating:** 6
**Confidence:** 2

**Summary:**

Authors propose a new Bayesian approach to derive and extend the federated Alternating Direction Method of Multipliers (ADMM). We show that the solutions of variational-Bayesian objectives are associated with a duality structure that not only resembles ADMM but also extends it， which opens a new direction to use Bayes to extend ADMM and other primal-dual methods.

**Strengths:**

Authors introduced a Bayesian duality, from which an extension of ADMM that optimizes over distributions naturally follows. For Gaussians with fixed variance, they recover regular ADMM and general Gaussians give Newton-like methods and IVON-ADMM. These show good performance when compared to recent baselines. Other approximating distributions may lead to new interesting splitting algorithms, and more generally, which opens up new research paths to extend and improve primal-dual algorithms using Bayesian ideas.

**Weaknesses:**

In the federated learning ADMM framework, there are theoretical guarantees for communication complexity and iterative complexity. Can the author briefly discuss the communication complexity and iteration complexity of Bayesian ADMM.

**Questions:**

In the federated learning ADMM framework, there are theoretical guarantees for communication complexity and iterative complexity. Can the author briefly discuss the communication complexity and iteration complexity of Bayesian ADMM.

---

> ### Author Response · Authors · 2025-11-21
> **Thank you for your review!**
>
> We thank the reviewer for their time and positive evaluation of our work.
>
> > Q1: In the federated learning ADMM framework, there are theoretical guarantees for communication complexity and iterative complexity. Can the author briefly discuss the communication complexity and iteration complexity of Bayesian ADMM.
>
> A1: We discuss communication cost and complexity in lines 357-363. To be clear, the computation complexity is the same as other methods (for example, FedDyn), and we have made this clearer in the paper (line 360). We show that BayesADMM converges for specific losses and exponential families in Proposition 3.2, but further theoretical guarantees for BayesADMM are an open problem in optimization and out-of-scope for this work.

---

### Official Review · Reviewer_B61X · 2025-11-10

**Soundness:** 2
**Presentation:** 2
**Contribution:** 3
**Rating:** 4
**Confidence:** 3

**Summary:**

The authors propose an analogous extension to federated ADMM in the context of variational inference. This provides an extension of ADMM-like federated procedures based on the duality of various exponential distributions.

**Strengths:**

The paper proposes a novel ADMM-like extension to federated learning, with good experimental results.

**Weaknesses:**

It seemed that the argument was more by analogy than exact equality. The claim that ADMM is recovered exactly is misleading because it requires an approximation and therefore is not necessarily recovered exactly.

A couple minor points:
- In equation 3, a subscript k is missing
- On line 195 A* should be defined in the main text rather than just in the appendix.
- In figure 4, it would be helpful to mention that each line is numbered with the iteration number.

**Questions:**

Why is it valid to just switch back and forth between variational inference and MLE? I was not convinced that equation 5 was equivalent to equation 1. Along this line, the notation in the paper blurred distinction between parameters and distributions (see equation 4 for example and line 976).
In equation 12-14, $\bar{\lambda}$ is not defined. Perhaps include an update equation for it. Is that because it is a deterministic mapping from $\bar{\mu}$? Similarly for $\bar{q}$, is the update from equation 4?
Under what circumstances does BayesADMM converge?

---

> ### Author Response · Authors · 2025-11-21
> **Thank you for your review!**
>
> We thank the reviewer for carefully reading our work and the constructive feedback, and are glad that they appreciate our experimental results. We have posted an updated version (changes in blue) which we believe addresses all comments and concerns. We would appreciate it if the reviewer will consider increasing their score based on this.
>
> > Q1: It seemed that the argument was more by analogy than exact equality. The claim that ADMM is recovered exactly is misleading because it requires an approximation and therefore is not necessarily recovered exactly.
>
> A1: Thanks for the suggestion. We have now modified the abstract and introduction (see the blue text in line 13 and 42) to clarify this issue. As we mention in the introduction (line 44-45), our claim is that our proposal can recover ADMM-like updates unlike the method of Swaroop et al. (2025). This is the main problem we are addressing. However, we are also able to recover the classical ADMM if we make an additional approximation using the delta method (see line 42-43).
>
> > Q2: Why is it valid to just switch back and forth between variational inference and MLE? I was not convinced that equation 5 was equivalent to equation 1.
>
> A2: We do not claim that equation 5 is equivalent to equation 1, rather, equation 5 is the variational objective, and there are two equivalent forms of writing this objective (lines 172-173). We use a similar lifting procedure to go from ADMM to BayesADMM in the paper. We show that approximations to the algorithm targeting the lifted objective (BayesADMM) can recover methods targeting the original objective (ADMM) when expectations are approximated with a single sample at their mean. This approach is called the “delta method” and is a valid and well-known method, and discussed in Khan & Rue (2023).
>
> > Q3: Along this line, the notation in the paper blurred distinction between parameters and distributions (see equation 4 for example and line 976).
>
> A3: We are happy to improve the notation, but are unsure what the reviewer means. Parameters are bold-face and distributions are non-bold. Equation 4 is written as distributions, and when these distributions are exponential family distributions, we can also write these updates using their parameters (mu and lambda). We have updated Appendix H to not use the parameter \bold{t_k} to avoid confusion.
>
> > Q4: In equation 12-14, \bar lambda is not defined. Perhaps include an update equation for it. Is that because it is a deterministic mapping from \bar mu? Similarly for \bar q, is the update from equation 4?
>
> A4: We define \bar lambda in lines 230-231. It is the natural parameter of \bar q. \bar lambda, \bar mu and \bar q indeed are related through a deterministic mapping. \lambda is the natural parameter, \mu is the expectation parameter and q is the distribution.
>
> > Q5: Under what circumstances does BayesADMM converge?
>
> A5: A complete convergence analysis or guarantees for Bayesian algorithms (BayesADMM or PVI in Ashman et al. (2022) and Swaroop et al. (2025)) is a difficult open problem that would constitute a paper of its own. Unlike these previous works however, we make partial progress towards this and show in Proposition 3.2 that for certain losses and exponential families, then BayesADMM converges (and we show this empirically in Figure 3(b)). We also note that our method inherits convergence of regular ADMM when using isotropic Gaussians posteriors. Previous work such as Swaroop et al and Ashman et al do not have this property.
>
> > Q6: A couple minor points: [...]
>
> A6: Thanks for pointing these out. These are addressed in the updated version (all changes are highlighted in blue).

---

### Meta-Review · Area_Chair_3s3k · 2026-01-07

**Summary:**

The paper proposes a principled “Bayesian duality” view of variational Bayes that yields ADMM-like federated updates and, when instantiated with diagonal Gaussians + IVON, shows consistent empirical gains

**Reviewer Concerns:**

Reviewer B61X

(1) “Recovered exactly” claim misleading; seems more analogy than equality — Addresed: authors say they revised abstract/introduction to clarify “ADMM-like updates” and that exact recovery uses a "delta approximation"


(2) Clarity regarding VI and MLE... Eq. 5 vs Eq. 1 equivalence
 — Addressed: authors explicitly say they do not claim Eq. 5 ≡ Eq. 1 and explain the delta method / lifting viewpoint

(3) clarity+notation issues -- Authors provide some explanation but it is unclear to the authors and to the AC what the reviewer's exact confusion was about.

(4) Convergence analysis -- Authors acknowlege this is an open problem but provide foundation for some progressin this direction in this paper; also suggest that under gaussian isotropy the method still converges because of analogue to regular admm.

Reviewer NFZi

(1) Wants communication/iteration complexity discussion -- Authors address this partially with some discussion; this is beyond scope of this paper though.
.
Reviewer DtMH

(1) Formulation soundness issues -- derivation from Lagrangian unclear, dual update replacement seems heuristic which makes the duality angle fragile -- Largely addressed : authors say the method is not derived from the Lagrangian, remove it to avoid confusion, and claim validity via a saddle-point objective / fixed-point stationarity argument

(2) Experiments not conclusive, missing ablation studies.
 — Partially addressed: authors say Tables 1/2 provide quantitative advantages and give runtime/overhead statements; i didnt see additional ablation studies added.

(3) clarity/notational issues -- The authors concede they will fix.

Reviewer WVuS:

(1) Compute/comm complexity scaling ... Addressed: authors provide scaling and give a runtime statement for their setup .
(2) Error bars -- Authors say 3 seeds were used
(3) limited-communication budget tradeoff -- Authors respond those experiments exist.
.(4) Ablation study for hyper-parameter sensitivity -- Added by authors
(5) Minor clarity issues - addressed
(6) system heterogeneity robustness -- Not addressed

This work proposes a unifying “duality” view and yields practical algorithm variants with reported empirical gains; the authors also added comparisons (PVI/PVI+IVON, FedLap, etc.) and clarified core claims .

**Reviewer Scores:**

I believe concerns of WVuS and DtMh were adequately addressed. DtMH should raise the score. B61X should also be reasonably satisfied provided they agree that the convergecne analysis is a very hard open problem.

---

### Decision · Program_Chairs · 2026-01-26

Accept (Poster)